# Global Geographic Distribution and Host Range of *Fusarium circinatum*, the Causal Agent of Pine Pitch Canker

Rein Drenkhan [1,*], Beccy Ganley [2], Jorge Martín-García [3,4], Petr Vahalík [5], Kalev Adamson[1], Katarína Adamčíková [6], Rodrigo Ahumada [7], Lior Blank [8], Helena Bragança [9], Paolo Capretti [10], Michelle Cleary [11], Carolina Cornejo [12], Kateryna Davydenko [13,14], Julio J. Diez [3,4], Hatice Tuğba Doğmuş Lehtijärvi [15], Miloň Dvořák [5], Rasmus Enderle [16], Gerda Fourie [17], Margarita Georgieva [18], Luisa Ghelardini [10], Jarkko Hantula [19], Renaud Ioos [20], Eugenia Iturritxa [21], Loukas Kanetis [22], Natalia N. Karpun [23], András Koltay [24], Elena Landeras [25], Svetlana Markovskaja [26], Nebai Mesanza [21], Ivan Milenković [27,28], Dmitry L. Musolin [29], Konstantinos Nikolaou [30], Justyna A. Nowakowska [31], Nikica Ogris [32], Funda Oskay [33], Tomasz Oszako [34], Irena Papazova-Anakieva [35], Marius Paraschiv [36], Matias Pasquali [37], Francesco Pecori [38], Trond Rafoss [39], Kristina Raitelaitytė [26], Rosa Raposo [4,40], Cecile Robin [41], Carlos A. Rodas [42], Alberto Santini [38], Antonio V. Sanz-Ros [4,43], Andrey V. Selikhovkin [29,44], Alejandro Solla [45], Mirkka Soukainen [46], Nikoleta Soulioti [47], Emma T. Steenkamp [17], Panaghiotis Tsopelas [47], Aleksandar Vemić [27], Anna Maria Vettraino [48], Michael J. Wingfield [17], Stephen Woodward [49], Cristina Zamora-Ballesteros [50] and Martin S. Mullett [28,51]

1. Institute of Forestry and Rural Engineering, Estonian University of Life Sciences, Fr. R. Kreutzwaldi 5, 51006 Tartu, Estonia; rein.drenkhan@emu.ee (R.D.); kalev.adamson@emu.ee (K.A.)
2. The New Zealand Institute for Plant and Food Research Limited, 412 No.1 Road RD2, 3182 Te Puke, New Zealand; Beccy.Ganley@plantandfood.co.nz
3. Department of Plant Production and Forest Resources, University of Valladolid, Avenida de Madrid 44, 34071 Palencia, Spain; jorgemg@pvs.uva.es (J.M.-G.); jdcasero@pvs.uva.es (J.J.D.)
4. Sustainable Forest Management Research Institute, University of Valladolid – INIA, Avenida de Madrid 44, 34071 Palencia, Spain; jorgemg@pvs.uva.es (J.M.G.); raposo@inia.es (R.R.); asanzros@gmail.com (A.V.S.-R.)
5. Faculty of Forestry and Wood Technology, Mendel University in Brno, Zemědělská 3, 61300 Brno, Czech Republic; petr.vahalik@mendelu.cz (P.V.); milon.dvorak@mendelu.cz (M.D.)
6. Department of Plant Pathology and Mycology, Institute of Forest Ecology, Slovak Academy of Sciences, Akademická 2, 949 01 Nitra, Slovakia; katarina.adamcikova@ife.sk
7. Bioforest S.A., Km. 15 S/N, Camino a Coronel, 403 0000 Concepcion, Chile; rodrigo.ahumada@arauco.com
8. Department of Plant Pathology and Weed Research, Agricultural Research Organization, Volcani Center, HaMaccabim Rd 68, 7528809 Rishon LeZion, Israel; liorb@volcani.agri.gov.il
9. Instituto Nacional de Investigação Agrária e Veterinária, I.P. (INIAV, I.P.) & GREEN-IT Bioresources for Sustainability, ITQB NOVA. Av da República, Quinta do Marquês, 2780-159 Oeiras, Portugal; helena.braganca@iniav.pt
10. Department of Agriculture, Food, Environment and Forestry, Università degli Studi di Firenze, Piazzale delle Cascine, 18, 50144 Firenze, Italy; paolo.capretti@unifi.it (P.C.); luisa.ghelardini@unifi.it (L.G.)
11. Southern Swedish Forest Research Centre, Swedish University of Agricultural Sciences, Sundsvägen 3, 23053 Alnarp, Sweden; Michelle.Cleary@slu.se
12. WSL Swiss Federal Research Institute, Zürcherstrasse 111, 8903 Birmensdorf; Switzerland; carolina.cornejo@wsl.ch
13. Department of Forest Protection, G. M. Vysotskiy Ukrainian Research Institute of Forestry and Forest Melioration, 61024 Kharkiv, Ukraine; kateryna.davydenko@gmail.com
14. Department of Forest Mycology and Plant Pathology, Swedish University of Agricultural Sciences, 75007 Uppsala, Sweden
15. Faculty of Forestry, Isparta University of Applied Sciences, 32260 Isparta, Turkey; tugbadogmus@isparta.edu.tr
16. Institute for Plant Protection in Horticulture and Forests, Federal Research Centre for Cultivated Plants (JKI), Messeweg 11/12, 38104 Braunschweig, Germany; rasmus.enderle@julius-kuehn.de

[17] Department of Biochemistry, Genetics and Microbiology, Forestry and Agricultural Biotechnology Institute (FABI), University of Pretoria, Lynwoond and University Roads, 0028 Pretoria, South Africa; gerda1.fourie@fabi.up.ac.za (G.F.); emma.steenkamp@fabi.up.ac.za (E.T.S.); mike.wingfield@fabi.up.ac.za (M.J.W.)

[18] Department of Forest Entomology, Phytopathology and Game fauna, Forest Research Institute, Bulgarian Academy of Sciences, 132 St. Kliment Ohridski Blvd, 1756 Sofia, Bulgaria; margaritageorgiev@gmail.com

[19] Department of Natural Resources, Natural Resources Institute Finland (Luke), Latokartanonkaari 9, 00790 Helsinki, Finland; jarkko.hantula@luke.fi

[20] ANSES Plant Health Laboratory, Unit of Mycology, Domaine de Pixérécourt, Bât. E., 54220 Malzéville, France; renaud.ioos@anses.fr

[21] Forestry Science Department, Neiker Institute, Campus Agroalimentario de Arkaute, S/N 01080 Arkaute, Álava, Spain; eiturritxa@neiker.eus (E.I.); nmesanza@neiker.eus (N.M.)

[22] Department of Agricultural Sciences, Biotechnology & Food Science, Cyprus University of Technology, Arch. Kyprianos Str. 30, 3603 Limassol, Cyprus; loukas.kanetis@cut.ac.cy

[23] Russian Research Institute of Floriculture and Subtropical Crops, Jānis Fabriciuss str., 2/28, 354002 Sochi, Russia; nkolem@mail.ru

[24] Forest Protection Department, NARIC Forest Research Institute, Hegyalja u. 18, 3232 Mátrafüred, Hungary; koltaya@erti.hu

[25] Laboratorio de Sanidad Vegetal, Gobierno del Principado de Asturias, C/ Lucas Rodríguez Pire, 4-bajo, 33011 Oviedo, Spain; elena.landerasrodriguez@asturias.org

[26] Institute of Botany, Nature Research Centre, Žaliųjų ežerų 49, 08412 Vilnius, Lithuania; svetlana.markovskaja@gamtc.lt (S.M.); kristina.raitelaityte@gmail.com (K.R.)

[27] Department of Forest Protection, Faculty of Forestry, University of Belgrade, Kneza Višeslava 1, 11030 Belgrade, Serbia; ivan.milenkovic@sfb.bg.ac.rs (I.M.); aleksandar.vemic2@gmail.com (A.V.)

[28] Phytophthora Research Centre, Mendel University in Brno, Zemědělská 3, 61300 Brno, Czech Republic; Martin.mullett@mendelu.cz

[29] Department of Forest Protection, Wood Science and Game Management, Saint Petersburg State Forest Technical University, Institutskiy per., 5, 194021 Saint Petersburg, Russia; musolin@gmail.com (D.L.M.); a.selikhovkin@mail.ru (A.V.S.)

[30] Department of Forests, Ministry of Agriculture, Rural Development and Environment, Loukis Akritas 26, 1414 Nicosia, Cyprus; knikolaou@fd.moa.gov.cy

[31] Institute of Biological Sciences, Faculty of Biology and Environmental Sciences, Cardinal Stefan Wyszynski University in Warsaw, Wóycickiego 1/3 Street, 01-938 Warsaw, Poland; j.nowakowska@uksw.edu.pl

[32] Department of Forest Protection, Slovenian Forestry Institute, Večna pot 2, 1000 Ljubljana, Slovenia; nikica.ogris@gozdis.si

[33] Faculty of Forestry, Çankırı Karatekin University, 18200 Çankırı, Turkey; fundaoskay@karatekin.edu.tr

[34] Department of Forest Protection, Forest Research Institute in Sękocin Stary, Braci Leśnej 3, 05-090 Raszyn, Poland; t.oszako@ibles.waw.pl

[35] Faculty of Forestry, Ss. Cyril and Methodius University in Skopje, 16 Makedonska brigada br.1, 1000 Skopje, Republic of North Macedonia; ipapazova@sf.ukim.edu.mk

[36] Department of Forest Protection, National Institute for Research and Development in Forestry – Brașov Station, Cloşca 13, 500040 Brașov, Romania; marius.paraschiv@icas.ro

[37] Department of Food, Environmental and Nutritional Sciences, University of Milan, Via Celoria 2, 20133 Milano, Italy; matias.pasquali@unimi.it

[38] Institute for Sustainable Plant Protection – C.N.R., Via Madonna del Piano, 10, 50019 Sesto Fiorentino, Italy; alberto.santini@cnr.it (A.S.); francesco.pecori@ipsp.cnr.it (F.P.)

[39] Biotechnology and Plant Health Division, Norwegian Institute of Bioeconomy Research, 1431 Ås, Norway; trond.rafoss@gmail.com

[40] Centre of Forest Research, National Institute for Agricultural and Food Research and Technology (INIA), C. Coruna, 28040 Madrid, Spain; raposo@inia.es

[41] INRAE, Univ. Bordeaux, BIOGECO, F-33610 Cestas, France; cecile.robin@inrae.fr

[42] Forest Health Protection Programme, Smurfit Kappa Colombia – University of Pretoria, Calle 15 N° 18 - 109. Yumbo – Valle Colombia, 760502 Cali, Colombia; Carlos.rodas@smurfitkappa.com.co

[43] Plant Pathology Laboratory, Calabazanos Forest Health Center (Regional Government of Castilla y León), Polígono Industrial de Villamuriel S/N, 34190 Villamuriel de Cerrato, Palencia, Spain; asanzros@gmail.com

[44] Department of Biogeography and Environmental Protection, Saint Petersburg State University, Universitetskaya emb., 13B, 199034 Saint Petersburg, Russia

[45] Faculty of Forestry, University of Extremadura, Avenida Virgen del Puerto 2, 10600 Plasencia, Spain; asolla@unex.es

[46] Laboratory and Research Division, Plant Analytics Unit, Finnish Food Authority, Mustialankatu 3, 00790 Helsinki, Finland; mirkka.soukainen@ruokavirasto.fi

[47] Institute of Mediterranean Forest Ecosystems, Terma Alkmanos, 11528 Athens, Greece; soulioti@fria.gr (N.S.); tsop@fria.gr (P.T.)

[48] Department for Innovation in Biological, Agro-food and Forest Systems (DIBAF), University of Tuscia, Via S. Camillo de Lellis, 01100 Viterbo, Italy; vettrain@unitus.it

[49] School of Biological Sciences, University of Aberdeen, Cruickshank Building, St. Machar Drive, Aberdeen AB24 3UU, UK; s.woodward@abdn.ac.uk

[50] Department of Vegetal Production and Forestry Resources, College of Agricultural and Forestry Engineering, University of Valladolid, Av Madrid 44, 34004 Palencia, Spain; cristinazamoraballesteros@gmail.es

[51] Forest Research Forest Research, Alice Holt Lodge, Surrey, Farnham GU10 4LH, UK; martinmullett@hotmail.com

**\*** Correspondence: rein.drenkhan@emu.ee

**Abstract:** *Fusarium circinatum*, the causal agent of pine pitch canker (PPC), is currently one of the most important threats of *Pinus* spp. globally. This pathogen is known in many pine-growing regions, including natural and planted forests, and can affect all life stages of trees, from emerging seedlings to mature trees. Despite the importance of PPC, the global distribution of *F. circinatum* is poorly documented, and this problem is also true of the hosts within countries that are affected. The aim of this study was to review the global distribution of *F. circinatum*, with a particular focus on Europe. We considered (1) the current and historical pathogen records, both positive and negative, based on confirmed reports from Europe and globally; (2) the genetic diversity and population structure of the pathogen; (3) the current distribution of PPC in Europe, comparing published models of predicted disease distribution; and (4) host susceptibility by reviewing literature and generating a comprehensive list of known hosts for the fungus. These data were collated from 41 countries and used to compile a specially constructed geo-database (http://bit.do/phytoportal). A review of 6297 observation records showed that *F. circinatum* and the symptoms it causes on conifers occurred in 14 countries, including four in Europe, and is absent in 28 countries. Field observations and experimental data from 138 host species revealed 106 susceptible host species including 85 *Pinus* species, 6 non-pine tree species and 15 grass and herb species. Our data confirm that susceptibility to *F. circinatum* varies between different host species, tree ages and environmental characteristics. Knowledge on the geographic distribution, host range and the relative susceptibility of different hosts is essential for disease management, mitigation and containment strategies. The findings reported in this review will support countries that are currently free of *F. circinatum* in implementing effective procedures and restrictions and prevent further spread of the pathogen.

**Keywords:** invasive pathogen; climate change; interactive map of pathogen; susceptibility

## 1. Introduction

*Fusarium circinatum* (teleomorph *Gibberella circinata* Nirenberg and O'Donnell [1]) is an invasive pathogen that causes a disease known as pine pitch canker (PPC). This fungus is a quarantine organism, included in the EPPO (European and Mediterranean Plant Protection Organization) A2 list and regulated in the EU (European Union) [2]. In nurseries and the wider environment, the pathogen affects pines (*Pinus* spp.) and Douglas-fir (*Pseudotsuga menziesii*) [3,4]. It has also been isolated from asymptomatic plants (Poaceae, Asteraceae, Lamiaceae, Rosaceae) growing close to PPC-affected trees in pine stands [5–7]. Additionally, artificial inoculation trials have shown the potential for *F. circinatum* to infect a wide range of plant genera, e.g., *Abies*, *Larix*, *Libocedrus*, *Picea* [3,8–10], although natural infections of species in these taxa have not been reported.

*Fusarium circinatum* can affect all stages of pine development. Being seed-borne [11], it can cause seed and seedling mortality (pre- and post-emergence damping-off, respectively), and lignified seedling decay (late damping-off) [12]. The pathogen also causes dieback of branches and stems on young and mature trees where the main symptoms are copious resin ('pitch') production from cankers, hence the common name "pine pitch canker disease" [13–15]. Infection is usually via wounds through which spores gain entry into the plant tissue [16,17]. However, wounds are not always necessary for infection although they are for disease development [18]. The dispersal of the infective propagules occurs via agents such as insects, water, and wind [19–21]. However, the main avenues for long-distance movement of the fungus are associated with human activities, particularly plant trade and movement of contaminated soil and equipment [22,23].

Pine pitch canker was first described in the Southeast USA (North and South Carolina) in 1945 [24], where outbreaks tended to occur in poorly managed stands or after severe drought events [17]. Since then, *F. circinatum* has been recorded in Africa, Asia, South America, and Southern Europe, although information concerning its distribution in these regions is often not easily accessible or uniformly presented [25–27]. However, the pathogen is now known from most pine growing regions, generally with high incidence in Mediterranean and sub-tropical climates and some spread into temperate regions [28]. Because its spread and establishment is strongly dependent on climatic conditions, primarily temperature and humidity [4,28,29], *F. circinatum* is unlikely to spread to cooler, northern latitudes despite the presence of susceptible hosts in these areas [29,30]. Nevertheless, global trade has exacerbated the spread of many forest pathogens, and this trend seems set to continue [31,32]. Introduction of the pathogen via anthropogenic activities into nurseries or areas with suitable microclimates could lead to disease spread into what have hitherto been thought of as generally less suitable areas.

The European COST Action FP1406 "Pine pitch canker – strategies for management of *Gibberella circinata* in greenhouses and forests (PINESTRENGTH)" brought together scientists and stakeholders from 36 countries to establish a European-focused network dedicated to increasing our understanding of *F. circinatum* and its effects on pine. The main objectives were to increase knowledge on the biology, ecology, and spread pathways of *F. circinatum*; to evaluate the potential to develop effective and environmentally friendly prevention and mitigation strategies and to deliver these outcomes to stakeholders and policy makers. In this regard, updated information on the geographic distribution and host range of the pathogen, as well as on the relative susceptibility of different hosts, were considered. These factors represent important elements of disease management, mitigation and containment strategies. This would potentially also allow countries currently free of the pathogen to implement effective procedures and restrictions to prevent its introduction.

In this review, we considered the global distribution of *F. circinatum*, with a particular focus on Europe. More specifically the objectives were to (1) determine presently available and historical pathogen records, based on the confirmed reports from Europe and globally, (2) review the global populations and genetic diversity of the pathogen, (3) compare the current distribution of PPC in Europe with published models of predicted disease distribution; and (4) provide a comprehensive and up to date list of susceptible hosts.

## 2. The Geographic Distribution of *F. circinatum*

The occurrence of *F. circinatum* is well known for some countries, while information regarding its distribution globally or within many countries is scattered or poorly documented, and in some cases records are erroneous. To present the current distribution of *F. circinatum*, a geo-database for the pathogen was developed and used to generate an interactive map (see Supplementary Materials and interactive map: http://bit.do/phytoportal).

The geo-database contains geographic coordinates of 6297 sampling or observation records from 106 different hosts in 41 countries (including states): Africa (1 country), North and South America (7 countries), Asia (5 countries, including the Asian part of the Russian Federation and the Asian part of Turkey), Europe (28 countries including the European part of the Russian Federation and the

European part of Turkey), and Oceania (2 countries). The interactive map shows the presence and first reports of *F. circinatum* in 14 countries, including four in Europe (Figure 1; Table 1).

In 12 countries (Brazil, Chile, Colombia, France, Italy, Japan, Mexico, Portugal, Spain, South Africa, Uruguay, and USA [not all states]), the presence of *F. circinatum* was confirmed using molecular methods, and in two countries (South Korea and Haiti) the presence of the pathogen was verified using classical morphological approaches (e.g., vegetative and reproductive traits). In France and Italy, *F. circinatum* has been found in nurseries and at public gardens, and in both of these European countries the pathogen is considered officially eradicated [27] (Figure 2; Table 1). PPC was considered absent in 28 countries (24 European countries, Australia, New Zealand, Turkey and Israel) after rigorous field observations and/or laboratory testing (see http://bit.do/phytoportal). Countries for which no data on presence or absence were available were not considered to be positive or negative. The data were obtained as described in the instructions of the geo-database for *F. circinatum* distribution (see **Supplementary Materials, Table S1**). A summary of pathogen distribution by continent is presented below.

*2.1. Europe*

In Europe, *F. circinatum* has been reported in four countries: Spain [33,34], Italy [35], France [36], and Portugal [37]. The first record of *F. circinatum* in Europe was in 1995 on nursery seedlings of *P. radiata* and *P. halepensis* in Galicia, northern Spain [33]. In 1997, the pathogen was evidently found in a nursery in the Basque Country, northern Spain, causing mortality of *P. radiata* seedlings [38–40], but *F. circiantum* presence was formally identified in 2004 [34]. The disease reappeared in northern Spain, in Asturias, some years later (2003–2004) on nursery seedlings of *P. radiata* and *P. pinaster* [34]. In 2004, the pathogen was reported for the first time in the forest environment, where it caused PPC of *P. radiata* in a 20-year-old forest plantation in Cantabria, northern Spain [34]. Thus, *F. circinatum* has been present for over 20 years in Spanish nurseries and over 10 years in forests.

In 2006, an eradication and control programme was launched to limit its spread in Spain, and in 2007 the EU adopted measures to prevent its spread to other member states [2]. Measures undertaken included the elimination of infected material and the establishment of an intensive and continuous monitoring programme in forests and of plant reproductive material from both public and private entities. In Castilla y León, *F. circinatum* was found from 2005 to 2013, both in nurseries and forest stands, but there have been no subsequent reports of the fungus in that region (Forest Health Service, direct communication). However, the pathogen remains active in several coastal areas despite eradication attempts. If *F. circinatum* cannot be eradicated from these regions, it is likely that new infections will occur and the pathogen will continue to spread to inland areas [41].

In Portugal, *F. circinatum* was first detected in 2007 from infected seedlings of *P. radiata* and *P. pinaster* in a nursery located in the centre of the country [37]. As a consequence of this first report, following EU and national rules, an action plan was implemented by the Forest Authority to establish extraordinary phytosanitary measures, aiming to eradicate and/or control the disease. In both Portugal and Spain, after each detection of *F. circinatum*, an infested zone and a buffer zone (at least 1 km wide) were established around the infested site. In Portugal, the survey and programme results (Figure 1, http://bit.do/phytoportal) showed that until 2016 all positive reports of *F. circinatum* were obtained exclusively from nurseries. In 2016, the fungus was also detected for the first time in a plantation of *P. radiata* in Minho province [42] and in the same province on two *P. pinaster* trees in 2018 [42]. In both cases, *Pinus* plants in nurseries and in forests were destroyed and the surrounding area intensively surveyed with no further positive findings to date [27]. Although the lack of new positive records suggests that the pathogen has been successfully eliminated, it is premature to declare official eradication in the whole country.

In Italy and France, *F. circinatum* appears to have been eradicated successfully. Pine pitch canker was reported in Italy in 2005 on ornamental plantings (Foggia, southern Italy) of the native species *P. pinea* and *P. halepensis* [35]. Infected plants were promptly removed and destroyed, and no new records of the disease have subsequently been reported in gardens, nurseries or the wider environment. In France, *F. circinatum* was recorded for the first time in 2005 in a private garden

(Perpignany, South France) on a declining Douglas-fir tree and a few declining pine trees [36] and was considered officially eradicated in 2008 after intensive monitoring [43,44]. In 2009, the fungus was found on *P. radiata* seedlings in two French nurseries: all infected plants and plants from the same nursery beds were removed and destroyed [45]. After two years of intensive survey in and around the nurseries, the pathogen was considered eradicated [27].

The current study gathered 6297 observations from 28 European countries (http://bit.do/phytoportal). In 24 of the 28 monitored European countries, there was no evidence of *F. circinatum* presence (i.e., all surveys and samples were negative) (Figures 1 and 2). Both morphological and molecular methods (species-specific PCR [46] or sequencing) for *F. circinatum* detection were used to determine presence or absence in 18 countries. In nine countries, only visual inspection of symptoms in the field or morphological diagnosis of cultures in the laboratory was used to confirm pathogen absence or presence. However, visual inspection alone may not be sufficiently reliable for *F. circinatum* detection and identification because the fungus may behave as an endophyte or latent pathogen with no visible external symptoms or it can be mistaken for other pathogens that cause similar symptoms [7,47,48]. It is preferable, therefore, to combine visual surveys with molecular detection methods for reliable and more precise identification [46,49,50]. In the current study, both visual and molecular surveys were considered.

### 2.2. North America and South America

*Fusarium circinatum* was first recorded on pines in southeastern North America in 1945 [24]. The pathogen was described from *Pinus virginiana* in North and South Carolina [24]. Pine pitch canker has subsequently been reported from other states including Alabama, Arkansas, California, Florida, Georgia, Indiana, Louisiana, Mississippi, Tennessee, Texas, and Virginia [26,29] (see Figures 1 and 2, Table 1). In Mexico, *F. circinatum* was recorded for the first time in 1989 on planted *P. halepensis* and natural stands of *P. douglasiana* and *P. leiophylla* [39]. Consistent with the idea that *F. circinatum* is native to Mexico [51,52], the pathogen is widespread in this country with records from at least nine states (Sinaloa, Nayarit, Mexico, Nuevo Leon, Puebla, Michoacan, Jalisco, Durango, and Tamaulipas). There were no published records for the pathogen in Canada (Tod Ramsfield, personal comm.) or in the USA states of Massachusetts and Washington [27].

The first report of PPC in Haiti was in 1953 [53], although thereafter no new information is available about the disease in that country. The first report of *F. circinatum* in South America was in *P. radiata* mother plants (hedges) in nurseries of Chile in 2001 [54]. Since then, the pathogen has been found in Uruguay, Colombia, and Brazil [55–58]. In Chile, Uruguay, and Brazil, the pathogen has been reported only in nurseries and it has apparently not spread to the forest environment. Conversely, in Colombia, *F. circinatum* was first detected in 2005, affecting seedlings of *P. patula*, *P. maximinoi,* and *P. tecunumanii* in nursery, but was later also found in isolated trees on plantations [56,57]. More recently in 2017, the pathogen was reported and identified as *F. circinatum* causing damages in *P. patula* and high elevation plantations of *P. tecunumanii* (Carlos Rodas, unpublished).

### 2.3. Asia

In Asia, *F. circinatum* is known to be present only in Japan and South Korea [59,60]. There are no records of the pathogen in the Russian Far East, nor in western Asia (e.g., Israel and Turkey, including the European part of Turkey) (see Figures 1 and 2, Table 1). In Japan, PPC was first recorded in 1981 on native *P. luchensis* trees on Amami Ōshima and Okinawa Islands [60]. In South Korea, PPC was reported from natural stands and plantations of *P. rigida* in the mid-1990s where it caused tree mortality in Seoul and Kangwon Provinces [59].

### 2.4. Africa

In Africa, *F. circinatum* has been reported only from South Africa, where it was documented for the first time in 1991 [12]. The pathogen was responsible for an outbreak of root and root collar disease on *P. patula* seedlings in a single nursery, and has subsequently spread to most pine seedling

production nurseries in the country [4]. Accordingly, various management strategies have been investigated and developed to limit the occurrence and spread of *F. circinatum* in commercial forestry, e.g., nursery hygiene practices to limit the build-up of inoculum [61–63] and the use of chemical and biologically derived compounds to boost plant defence responses [64]. However, *F. circinatum* remains a major challenge to seedling production and plantation establishment in South Africa [65,66].

In 2005, PPC was detected for the first time outside the nursery environment on established trees in a plantation of *P. radiata* and it is now commonly found on this species in the Western Cape Province in South Africa [67,68]. The pathogen has since been detected also in plantations of *P. greggii* in the Eastern Cape and KwaZulu-Natal Provinces where localized outbreaks of PPC have occurred [66,68,69]. Additionally, in the summer rainfall area of the country, localised outbreaks of PPC have been recorded in plantations of *P. patula*, which is the most widely planted *Pinus* species in South Africa [70]; Steenkamp and Wingfield, unpublished]. To limit losses related to PPC, considerable effort has been dedicated to develop and deploy planting stock that is tolerant or more resistant to PPC. These include less susceptible families of *P. patula* [71–73] and certain families of *P. maximinoi*, *P. pseudostrobus*, low-elevation *P. tecunumanii*, and *P. elliottii* var. *elliottii* [65,74]. Various hybrids have also been evaluated, with low-elevation *P. tecunumanii* x *patula*, *P. elliottii* x *caribaea*, and *P. patula* x *oocarpa* showing low levels of susceptibility to infection by *F. circinatum* [75,76], and many of these hybrids have already been deployed commercially.

## 2.5. Oceania

*Fusarium circinatum* has not been found in Oceania. In both Australia and New Zealand, surveillance programmes regularly monitor pine and Douglas-fir seedlings and mature trees for unwanted organisms including *F. circinatum.* Suspect samples are tested using morphological and molecular methods and, to date, all samples tested have proven negative for *F. circinatum* (see Figures 1 and 2). Both countries have strict border biosecurity regulations, and at least one potential introduction of the pathogen has been prevented. In this case, *F. circinatum* was detected in 2004 on scions of Douglas-fir from California, and the pre-border detection required the infected material to be destroyed before it was imported into New Zealand [77].

**Table 1.** Geographic distribution, by country, of *Fusarium circinatum* (FC) including the type of planting, year found, host species affected, and the method used to identify the pathogen.

| Continent/ Country/ State | Year of First Record of FC in Nursery and/or Wider Environment | Host Species | Type of Planting / Sampling Site | Identification Method | References or Data Holder |
| --- | --- | --- | --- | --- | --- |
| **Europe** | | | | | |
| France | **2005** [1, 2] | *Pseudotsuga menziesii, Pinus* sp. | Urban greenery | species-specific PCR, sequence analysis | [37] |
| France | 2008 [2] | *P. menziesii* | Nursery | species-specific PCR, sequence analysis | [44] |
| Italy | **2005** [1, 2] | *P. halepensis, P. pinea* | Urban greenery | species-specific PCR | [36] |
| Portugal | **2007** [1, 2] | *Pinus radiata, P. pinaster* | Nursery | species-specific PCR, sequence analysis | [38] |
| Portugal | 2016 [2] | *P. radiata* | Forest plantation | species-specific PCR, sequence analysis | [42] |
| Spain | **1995** [1] | *P. radiata, P. halepensis* | Nursery, in Galicia | visual observation | [34] |
| Spain | 1997 | *P. radiata* | Nurseries, in Basque Country | mycelial morphology | [39] |

| | | | | | |
|---|---|---|---|---|---|
| Spain | 2003 | *P. radiata, P. pinaster* | Nursery, in Asturias | species-specific PCR | [35] |
| Spain | 2004 | *P. radiata* | Plantation, in Cantabria | species-specific PCR | [35] |
| Spain | 2005 | *P. sylvestris, P. nigra, P. pinaster, P. pinea* | Nurseries in Castilla y León | morphological traits, species-specific PCR | Regional Forest Health Service |
| Spain | 2005 | *P. sylvestris, P. nigra, P. pinea, P. radiata* | Forest plantations, in Castilla y León | morphological traits, species-specific PCR | Regional Forest Health Service |
| **Asia** | | | | | |
| Japan | **1981**[1] | *P. luchuensis* | Forest, Amamioshima Island (Ryukyu Archipelago) and the Okinawa island | mycelial morphology | [60] |
| South Korea | **1995** [1] | *P. rigida* | Urban greenery, forest | mycelial morphology | [59] |
| **Africa** | | | | | |
| South Africa | **1991** [1] | *P. patula* | Nursery, Ngodwana Mpumalanga Province | mycelial morphology | [12] |
| South Africa | 2005 | *P. radiata* | Plantation, Tokai, Western Cape Province | species-specific PCR, sequence analysis | [67] |
| South Africa | 2007 | *P. greggii* | Plantation, Ugie, Eastern Cape Province | sequence analysis | [68,69] |
| South Africa | 2014 | *P. patula* | Plantation, Louis Trichardt, Limpopo Province | sequence analysis | [70] |
| **North America** | | | | | |
| Haiti | **1953** [1] | *P. occidentalis* | Natural forest | mycelial morphology | [53] |
| Mexico | **1989** [1] | *P. douglasiana, P. halepensis, P. leiophylla, P. greggii, P. patula* | Forest plantation and natural stand | unknown, probably visual observation | [40] |
| United States of America | | | | | |
| Alabama | 1980 | *Pinus taeda* | Forest plantation | mycelial morphology | [16] |
| Arkansas | 2013 | *P. elliottii, P. palustris, P. taeda* | unknown | unknown | [78] |
| California | 1986 | *P. radiata* | Forest plantation | visual observation | [79,80] |
| Florida | 1974 | *P. elliottii* var. *elliottii* | Forest plantation and seed orchards | unknown | [17,40 |
| Georgia | 1987 | unknown | Nursery | mycelial morphology | [81] |
| Georgia | 1988 | *P. taeda* | Forest plantation | visual observation | [82] |
| Indiana | 1994 | *P. elliottii* var. *elliottii* | Nursery | mycelial morphology | [83] |
| Louisiana | 2004 | *P. taeda* | Forest plantation | visual observation | [84] |
| Mississippi | 1974 | *P. taeda* | Seed orchards | visual observation | [17] |
| North Carolina | **1945** [1] | *P. virginiana, P. echinata, P. rigida* | Natural forest | visual observation, mycelial morphology | [24] |
| South Carolina | **1945** [1] | *P. virginiana* | unknown | visual observation, | [24] |

| | | | | mycelial morphology | |
|---|---|---|---|---|---|
| Tennessee | 1978 | *P. echinata* | unknown | visual observation | [85] |
| Texas | 1991 | unknown | Forest plantation | mycelial morphology | [80] |
| Virginia | 1985 | *P. echinata* | unknown | visual observation | [17,85] |
| **South America** | | | | | |
| Brazil | 2014 [1] | *P. taeda* | Nursery | sequence analysis | [58] |
| Chile | 2001 [1] | *P. radiata* | Nursery | sequence analysis | [54] |
| Colombia | 2005 [1] | *P. maximinoi, P. Patula, P. Tecunumanii* | Nursery | species-specific PCR, sequence analysis | [56,57] |
| Colombia | 2006 | *P. patula* | Forest plantation | sequence analysis | [56,57] |
| Uruguay | 2009 [1] | *P. taeda* | Nursery | species-specific PCR | [55] |

[1] Year (bold) of the first record of FC in the country; [2] Eradication procedures undertaken.

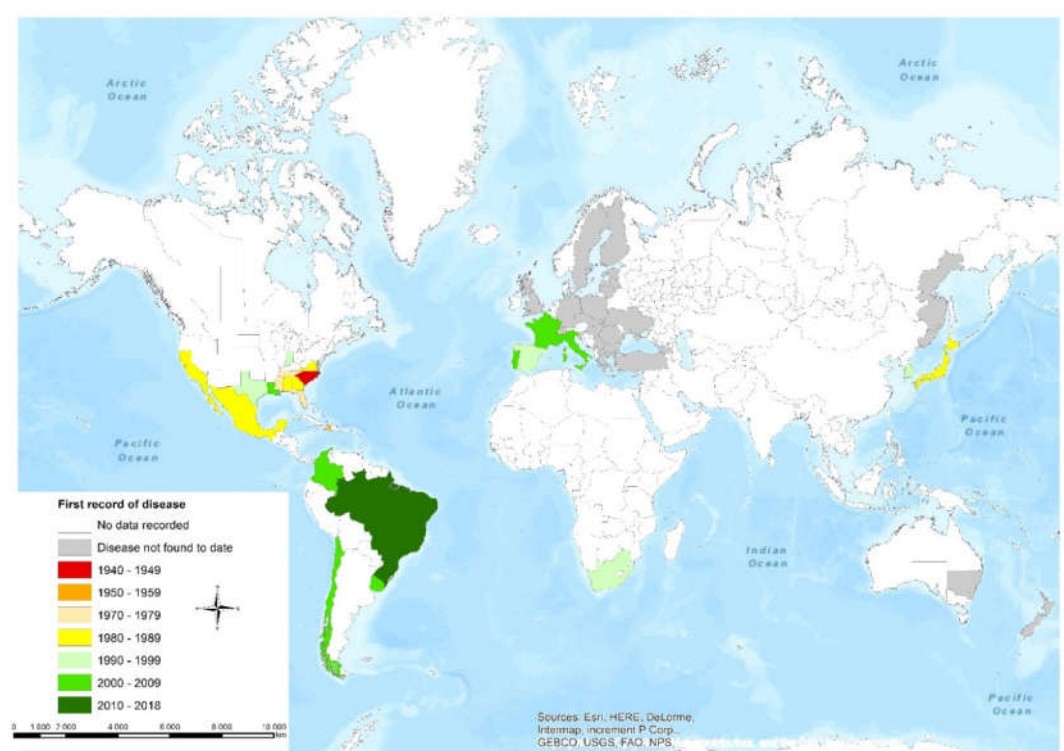

**Figure 1.** Historical dispersal of *Fusarium circinatum* according to the date the pathogen was first recorded in the country (see Table 1 for details). The data are based on literature and monitoring records.

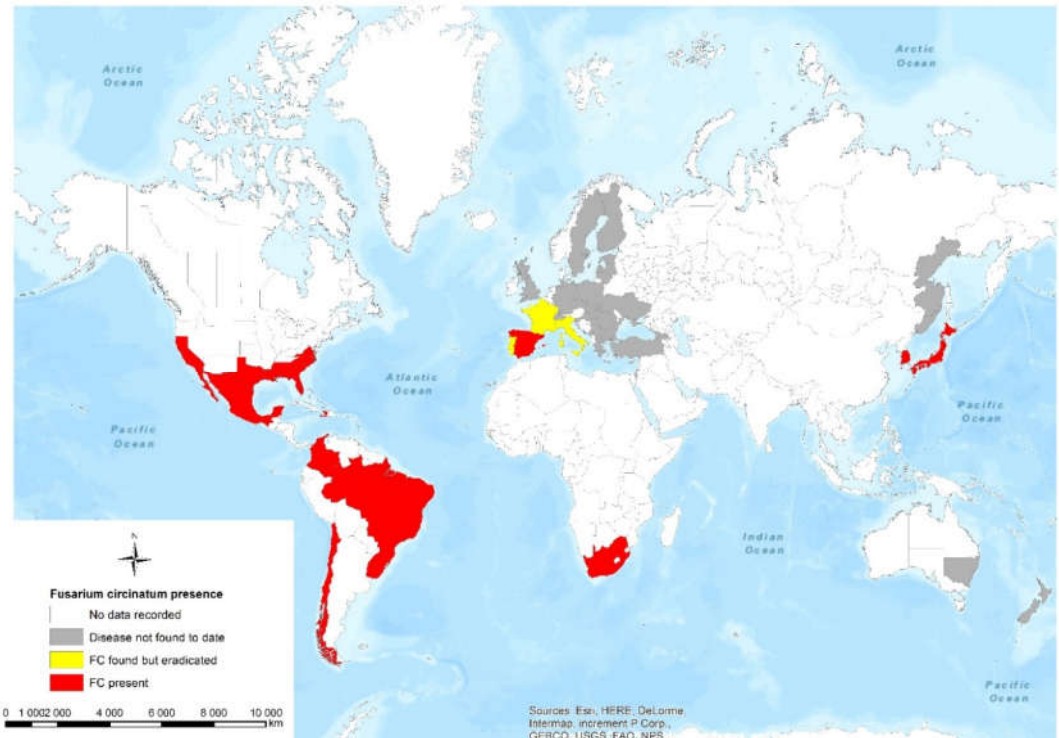

**Figure 2.** Global distribution of *Fusarium circinatum* showing where the pathogen is present, eradicated, not found or data are not available. See the interactive map: http://bit.do/phytoportal for detailed locations within countries.

### 3. Taxonomy and Evolution of *Fusarium circinatum*

The taxonomic history of *F. circinatum* is reasonably complex and the fungus has undergone numerous name changes since first discovery. The names *F. moniliforme* Sheldon var. *subglutinans* Wollenw. & Reinking, and *F. lateritium* (Nees) emend. Snyder and Hansen f. sp. *pini* Hepting were originally used [86–88]. The fungus was later designated as *F. subglutinans* (Wollenw. & Reinking) Nelson et al. f. sp. *pini* [87], and within *F. subglutinans*, it represented mating population H, which was one of several biological species contained within this asexual morph [89]. In 1998, data on the pathology, morphology and molecular evolution of the PPC fungus supported its formal recognition as the distinct species, *Gibberella circinata* Nirenberg & O'Donnell [90–92].

The PPC pathogen forms part of the *Gibberella fujikuroi* species complex, which broadly corresponds to section *Liseola* of *Fusarium* [91,93]. However, following the "One fungus, one name" convention [94], *Fusarium* is widely advocated as the sole name for the genus of fungi that includes some of the world's most important plant and animal pathogens and producers of mycotoxins [1]. Accordingly, the PPC pathogen is now commonly referred to as *F. circinatum*, while the broader clade to which it belongs is referred to as the *F. fujikuroi* species complex [1].

The genus *Fusarium* first emerged approximately 91.3 to 110 million years ago (Mya) in the middle Cretaceous, with the *F. fujikuroi* species complex emerging in the late Miocene, approximately 8.8 Mya [95]. A comprehensive phylogeographic treatment of the complex places *F. circinatum* firmly in the so-called American clade of the *F. fujikuroi* complex, and it is believed to have originated in Mexico/Central America [52,91,96]. A number of different lines of evidence support this theory, most crucially because *F. circinatum* is widespread in the region [97] where its populations are exceptionally diverse containing unique haplotypes [52]. *Fusarium circinatum* has recently also been found to be endophytic on *Zea mays*, another plant species with origins in Mexico/Central America [98,99]. The American clade also includes several other species originating from *Pinus* species (i.e., *F. marasasianum*, *F. parvisorum*, and *F. sororula*), some of which cause similar lesions on pines and behave

as aggressively as *F. circinatum* in virulence assays [100]. These data, together with the fact that Mexico/Central America has the greatest number of native *Pinus* species of any similar sized region in the world [101] and that this region represents the centre of origin for many of them [52,102], suggest that pines likely diversified alongside their *Fusarium* pathogens, including *F. circinatum*, in Mexico/Central America [91,100]. From here, *F. circinatum* has apparently been spread to other parts of the world via global trade associated with agriculture and forestry [4].

## 4. Population Dynamics of *Fusarium circinatum*

The population dynamics of *F. circinatum* are determined primarily by its reproductive mode. The fungus can reproduce both sexually and asexually, with each mode affecting the population structure differently. Asexual reproduction results in clonal populations, whereas sexual reproduction, involving meiosis and recombination, leads to the production of new genotypes and an increase in genetic diversity. Because *F. circinatum* is heterothallic, sexual reproduction requires the interaction of isolates of opposite mating type [89,103]. These mating types encode different sets of genes at the so-called mating type (*MAT1*) locus, which determines sexual interactions [104]. The two allelic versions of the *MAT1* locus are referred to as idiomorphs, and in *F. circinatum*, as in other Ascomycota, these are called *MAT1-1* and *MAT1-2* [89,105], each of which encodes a number of genes essential for completion of the sexual cycle [106].

In addition to mating type, another genetic factor that can influence sexual reproduction is female fertility [103,107]. Isolates of *F. circinatum* have clearly defined male and female roles, with female-fertile isolates able to form sexual fruiting structures as well as fertilize other female-fertile isolates. In contrast, male-only isolates can only fertilize female-fertile isolates and cannot produce sexual fruiting structures themselves. Because male-only strains are selected against during sexual recombination, those populations in which sexual recombination occurs have fewer male-only strains, while those that reproduce mainly asexually have more male-only strains [103,107,108]. Therefore, the ratio of female-fertile to male-only strains, as well as the ratio of *MAT1-1* to *MAT1-2* idiomorphs, can be used to inform whether a population is undergoing sexual recombination or is predominantly asexual [108]. Of the two characters, mating type is widely utilized in population studies of *F. circinatum* (e.g., [69,109,110]). By contrast, female fertility has been mainly utilized in studies of South African populations where asexual reproduction of the fungus seems to predominate [68,111,112].

More detailed information on population structure of pathogens such as *F. circinatum* can be obtained by investigating a number of different genetic loci at once. Vegetative compatibility groups (VCGs), which govern the formation of somatic heterokaryons, are controlled by multiple loci (i.e., vegetative incompatibility, or vic), and were the basis of the first methods applied to understand population diversity in *F. circinatum* [87,108]. Vegetative compatibility groups identify isolates that have the same allele at all or most of their *vic* loci [108,113], thereby providing a relatively simple means for studying subdivision in populations (e.g., [87,112,114]). Subsequently a range of DNA-based methods have also been applied in multilocus analyses of fungal populations. Those that have been developed specifically for *F. circinatum* includes sequence characterized amplified polymorphic markers [112], restriction-enzyme-based polymorphic DNA markers [52] and simple-sequence repeat (SSR) or microsatellite markers [115]. Most investigations of the population dynamics of *F. circinatum* typically use the aforementioned DNA-based methods and/or VCGs combined with mating type assays to determine the mode of reproduction and origin of the pathogen in a particular region, and in some cases for inferring potential introduction routes (see below).

In California and Japan, populations of *F. circinatum* have low levels of genetic diversity [52,87,109,116,117]. Although both mating types are present in California, sexual recombination appears to be extremely rare or absent. In Japan, where reproduction of the pathogen is asexual, only one mating type is known to be present [87,116,118,119]. These population traits are consistent with an introduced pathogen. Furthermore, the fact that both California and Japan share genotypes with the Southeastern USA strongly suggested that the pathogen came to these areas from the Southeastern USA [52,109].

In South Africa, diversity of *F. circinatum* is somewhat higher compared to populations in California and Japan. Both mating types of the fungus are present in the country, but one is typically under-represented in populations associated with PPC outbreaks in plantations [66,68–70]. This is different from the situation in Mexico and the Southeastern USA, where both mating types occur in high frequencies [52,70,109]. Comparison of the population associated with the first disease outbreak in the early 1990s with those from subsequent nursery outbreaks showed that diversity increased over time [112,114]. Although initially postulated to be due to sexual reproduction in a well-established population in the country, more recent evidence suggests that this increase in genetic diversity was due mostly to new introductions of the fungus into South Africa and not to sexual recombination [66,68–70]. Nevertheless, the initial nursery outbreak in South Africa was on *P. patula*, a species native to Mexico, which led to suggestions that the pathogen was imported from Mexico on infected seed [120]. The occurrence of a shared genotype between South Africa and Mexico, as well as the results of subsequent genetic clustering analyses, strongly supported this hypothesis [52,109].

In Chile, *F. circinatum* is currently present only in nurseries, and PPC on established trees is yet to be discovered. A shared genotype between Chilean and Mexican populations suggested Mexico as a potential source for the Chilean disease outbreak [109]. Conversely, the Uruguayan population of *F. circinatum* shares a genotype with the Southeastern USA, indicating this may be a separate introduction to South America and that the pathogen has not spread directly from Chile to Uruguay [109]. A more comprehensive analysis of the South American pathogen populations, including those from Brazil and Colombia, is required to elucidate their sources and transmission routes.

In Europe, the most comprehensive population analyses have been undertaken in Spain. In the Basque Country and Cantabria (Spain), the population comprises a single mating type (*MAT1-2*) and diversity is extremely low, consistent with a recently introduced, asexually reproducing pathogen [109,110]. More western populations in northern Spain (Galicia, Asturias, and Castilla y León) have marginally more diverse populations with both mating types present, yet no evidence of sexual recombination has been found [109]. In this country, the population of the fungus is structured in two distinct, well-differentiated groups, each dominated by a single genotype, which likely reflects two independent introductions of the pathogen [109]. Southeastern USA could be the source for both of these introductions due to populations from the two regions sharing genotypes [109]. Genotypes are also shared between Spain and Portugal and Spain and France, which included some of the most dominant genotypes in the region. Therefore, given the geographic proximity of these countries, the pathogen probably spread through these countries [109]. There is no information on the population biology and potential origin of *F. circinatum* in Italy, South Korea or Haiti.

The centre of origin of many tree pathogens and the specific source population leading to their introductions is commonly unknown [31,32]. Population genetic studies, however, have helped to elucidate possible pathways of introduction of *F. circinatum* globally. Nonetheless, some aspects remain unresolved, such as the relationships between the Southeastern USA and Mexican populations. In this case, it is unknown whether they are part of a continuous metapopulation spanning the entire region or whether the Southeastern USA population is separate but derived from that in Mexico [52]. However, knowledge regarding the centre of origin, source population and introduction pathways can help prevent further introductions by focusing quarantine measures and monitoring efforts where they are most effective (e.g., [121,122]). Similarly, the structure and reproductive mode of introduced populations can be used to establish management strategies, for example by helping to evaluate the risk of novel genotypes emerging that could potentially be more virulent or resistant to fungicides, or by targeting source areas of more resistant plant host material.

## 5. Climatic Influence on *Fusarium circinatum* Distribution and Modelling Potential Pathways of Introduction

Climate is a critical environmental determinant of the distribution of pathogens and a key driver of disease development [123–126]. As for most fungal pathogens, temperature and moisture are two of the most important climatic factors governing the distribution, spread, and symptom development of *F. circinatum*. For example, lesion lengths induced on pine by *F. circinatum* were positively

correlated with temperatures between 14 and 26 °C; no lesions developed on trees inoculated and maintained at 10 °C [127]. Occult precipitation (e.g., fog and mist) in coastal areas is considered the main reason PPC develops more rapidly and is more severe in *P. radiata* stands closer to the coast than inland [13,128,129]. This finding highlights the importance of moisture, which along with rain, also influences spore dissemination [14].

The climatic parameters suitable for *F. circinatum* infection, along with its known distribution range, have been used in CLIMEX modelling to estimate its potential distribution. Knowledge of the areas most suitable for disease development underpins PPC risk assessments and strategies to limit spread, control, and eradication. Ganley et al. [28] produced the first global CLIMEX model for PPC climate suitability, which provided a good fit with regions known to have the disease, particularly the Southeastern USA and Spain. Large areas of Southeast Asia and China were predicted to be optimal for the pathogen; as were Madagascar, Ethiopia, and equatorial regions of Africa, the North Island of New Zealand, certain coastal areas of Australia, many countries in Central America and large parts of South America. In Europe, the regions at greatest risk include wide areas of central and northern Portugal, northern and eastern Spain, south and coastal areas of France, coastal areas of Italy and the Balkans including Greece, Albania, Montenegro, Slovenia, and Croatia as well as north-western Turkey and western Georgia [28,29] (Figure 3). The model predicted that in Europe 690,000 km$^2$, or 7% of the total land area, were suitable (i.e., marginal, suitable or optimal ecoclimatic index) for disease development [130], with 578,135 km$^2$ considered optimal in the EU [29]. In these areas, pine forests (plantations and native forest) cover over 114,000 km$^2$, including areas with ornamental plantings [29]. A subsequent study [29], using higher resolution and more recent climatic data, showed a slightly broader area suitable for the disease, with optimal areas in the EU increasing to 682,387 km$^2$ and 813,612 km$^2$, respectively. Suitable areas in Europe are limited by cold stress at high altitudes and latitudes and by dry conditions in other parts of the EU [29].

A few studies have extended the CLIMEX modelling of suitable areas for PPC to incorporate models of climate change and spread from additional (i.e., theoretical future) introductions [30,130,131]. Globally, the area considered suitable for PPC decreased 39% to 58%, depending on the climate change scenario considered [130]. Suitable areas were projected to reduce in North America, South America, Asia, Africa, and Australia, although in Europe and New Zealand suitable areas increased under all climate change scenarios [130]. In Europe, the area considered suitable increased by 24 to 91% with the northern range extending as far north as the Netherlands or Denmark [130] and including southern Britain and Ireland by 2100 [30]. The predicted range extends and shifts northwards due to reduction in cold and drought stress [30,130]. In particular, an increase in summer and winter precipitation and temperatures in northern Europe (north of latitude 50° N) would make climatic conditions more favourable for *F. circinatum*. However, a detailed study in Spain indicated that the suitable area for PPC would likely decrease under future climate change scenarios in this country, with the suitable area condensing to a narrower coastal strip of north-central and western Spain by 2050 [131]. This reduction is probably a result of the predicted reduction in precipitation [131].

Möykkynen et al. [30] modelled the potential spread of *F. circinatum* in Europe and found that the fungus is likely to spread further through the pine forests of northern Spain (Galicia, Asturias, Cantabria, and Basque Country) and to southwest France (Aquitania), including some spread towards northern Portugal and southern Italy within the next 20 years. If new introductions to Central and North Europe occurred, Möykkynen et al. [30] predicted that *F. circinatum* could establish or spread to more northern parts of Europe. However, expansion would be limited due to the short dispersal distance of spores and the limited flight of insect vectors, although spread would be more likely through international trade, particularly via seed and nursery plants. Despite these predictions [29,30], there have been no records of *F. circinatum* in southwestern France, southern Italy or Greece to date (see http://bit.do/phytoportal). Nevertheless, continued vigilance is necessary in these areas given their climatic suitability and proximity to areas where the pathogen is known to be present. Studies that investigated climatic factors influencing infection and distribution of *F. circinatum* are in general in agreement that the main climatic constraints for global distribution are cold winter

temperatures and low precipitation during summer [28,130]. Modelling with high-resolution climatic data in Spain by Serra-Varela et al. [131] showed that the most relevant climatic variables for the distribution of *F. circinatum* are (i) temperature seasonality (annual range) (ii) minimum temperature during the coldest month, (iii) annual precipitation, and (iv) precipitation during the driest season.

The global distribution of *F. circinatum* obtained in this study (Figures 1 and 2) was summarized using a number of fine scale climatic and topological variables (summarized in Table 2). Only data points from the wider environment were analysed. Importantly, nursery records were not considered in the calculations because they include unnatural conditions of temperature and moisture and the likelihood that infected plants could be transported directly to a particular nursery from other areas. The analysis indicated that variation in monthly mean temperature and precipitation sum values is large for *F. circinatum* infested areas. Perhaps the most striking results were the mean temperature of the coldest months (minimum value) −14.2 °C in South Korea and highest temperature of warmest month (maximum value) +36.5 °C in Mexico (see Table 2; http://bit.do/phytoportal). Yet, the pathogen caused disease in natural and planted stands in Seoul, South Korea [59] and on native pines in Mexico [120]. This variation demonstrates the ability of *F. circinatum* to survive in its host and potentially cause disease across substantial temperature extremes, beyond the range of 14–26 °C that was positively correlated with lesions in artificial inoculation studies or the lower range temperature threshold of 10 °C for pathogen growth under laboratory conditions [127] and may hint towards ecological differences between strains of *F. circinatum* and thus indicate a need for better understanding of intraspecific variation within the species.

An overlap of the current European distribution of *F. circinatum* with the original CLIMEX parameters of Ganley et al. [28], but using newer high-resolution climate data [29] (Figures 3 and 4), shows that some of points fall in areas that were hitherto considered unsuitable for the pathogen. These mainly represented more recent points that were only included in the current study (nursery records were again not considered in the calculations). This finding suggests that the models could be improved in the light of new records of PPC in the last few years and updated with more recent climate data sets. The *F. circinatum* geo-database and web platform developed as part of this study aims to provide a suitable platform and resource to improve future modelling studies, as done for *Dothistroma* species [132]. Alternative explanations for the point discrepancies between modelled suitable climatic areas and newer PPC occurrences could be that the climatic conditions in these areas may have been uncharacteristically suitable for *F. circinatum* in recent years or even that the pathogen is adapting to a wider range of environmental conditions.

The climatic suitability models, models of disease spread, and summary of climatic variables described above relate to the wider, natural environment where PPC affects pine plantations or natural forests. However, *F. circinatum* is also a serious problem in nurseries, where it primarily causes pre- and post-emergence damping-off via seed and root infection. This manifestation of the pathogen is distinct from the disease expression of PPC on mature trees in natural forests and plantations. In nurseries, temperature and moisture conditions are often drastically different from the adjacent wider environment, particularly if seedlings are grown under protection (e.g., glasshouses or polytunnels). These conditions are often more suitable for disease development than is the case in the field, due to increased and stable moisture and temperature regimes. These conditions, together with importation of infected seeds and plants, can result in *F. circinatum* infections in nurseries far outside the range of generally suitable ecoclimatic conditions (e.g., positive findings in nurseries and forest stands in Castilla y León, Spain). The pathogen may well be able to thrive and cause considerable damage in nurseries in more northern latitudes, or areas generally not considered suitable for PPC. This possibly should be taken into account when assessing the risk to nurseries in these areas, and it is important to recognise that infected nursery plants represent a source of infection for trees in both forests and plantations.

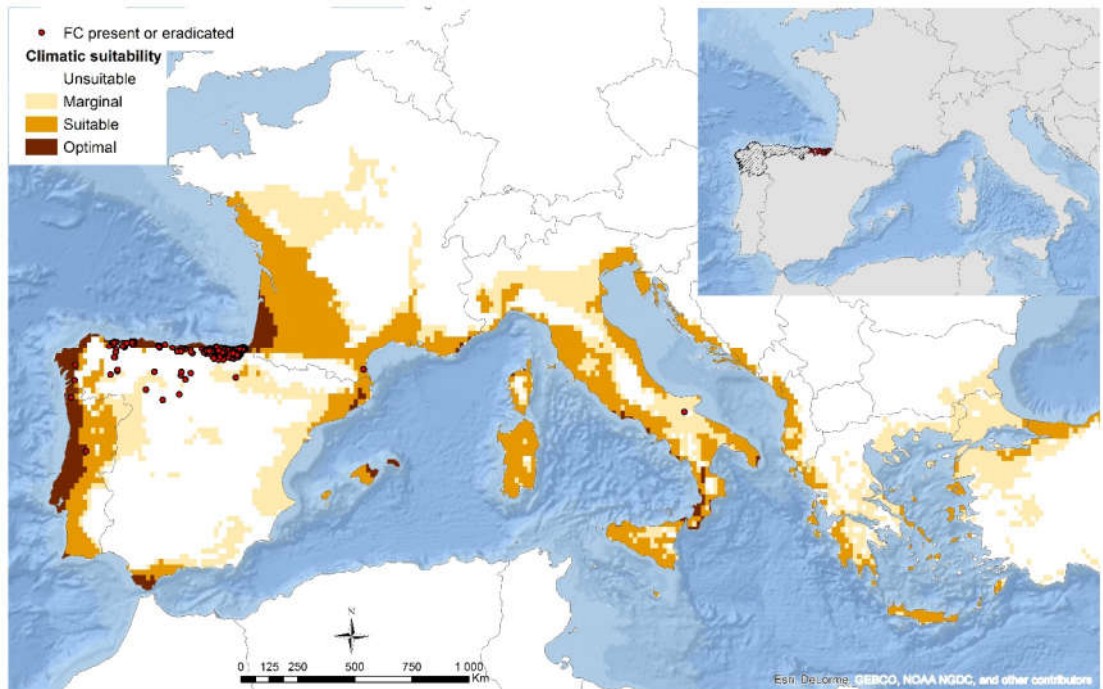

**Figure 3.** European climatic suitability for *Fusarium circinatum* based on the CLIMEX model parameters of Ganley et al. [28] using higher resolution climatic data [133]. The European distribution of non-nursery *F. circinatum* findings is shown as red dots in the main figure, while the inset displays the dataset used by Ganley et al. [28] in the original CLIMEX modelling.

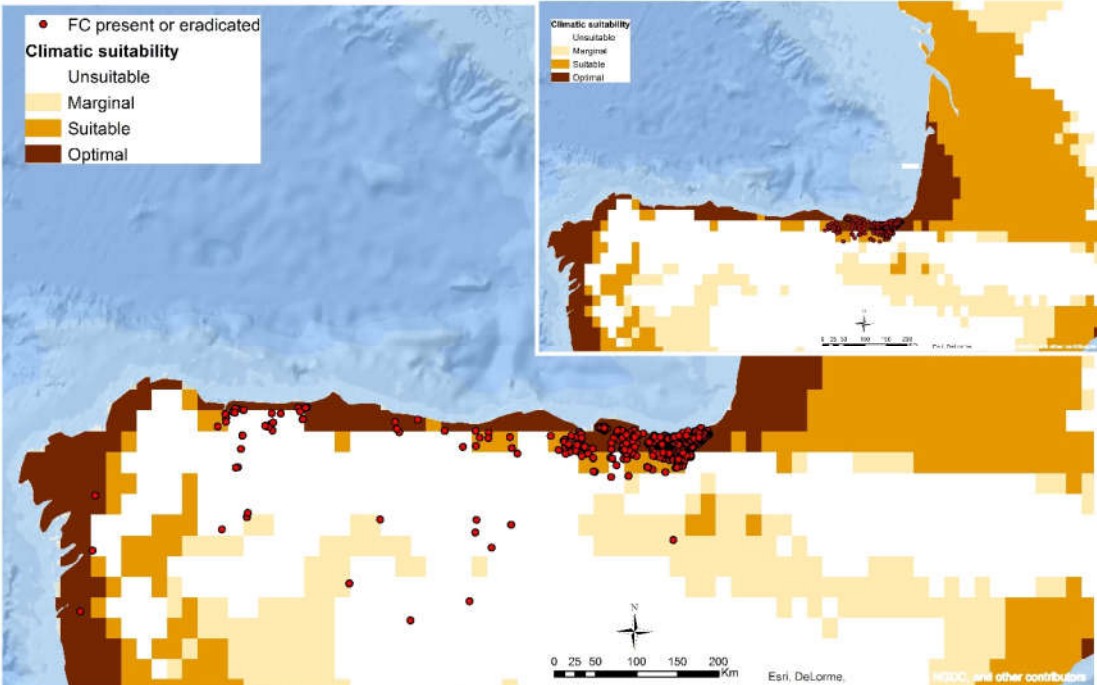

**Figure 4.** Climatic suitability for *Fusarium circinatum* based on the CLIMEX model parameters of Ganley et al. [28] using higher resolution climatic data [133] focused on the current European outbreak area. The European distribution of non-nursery *F. circinatum* findings is shown as red dots in the main

figure, while the inset displays the dataset used by Ganley et al. [28] in the original CLIMEX modelling.

**Table 2.** Minimum, average and maximum values of climatic and topographical variables from the dataset of Hijmans et al. [134] and new observations of current distribution of *Fusarium circinatum* from the geo-database presented in the current study.

|  | **Minimum** | **Average** | **Maximum** |
|---|---|---|---|
| Altitude | −3 m a.s.l. | 262 m a.s.l. | 3619 m a.s.l. |
| Annual mean temperature | 6.2 °C | 13.7°C | 25.4 °C |
| Mean temperature of the warmest month | 12.4 °C | 24.7 °C | 36.5 °C |
| Mean temperature of the coldest month | −14.2 °C | 4.8 °C | 16.7 °C |
| Annual precipitation sum | 324 mm | 1259 mm | 3062 mm |
| Precipitation sum of the wettest month | 51 mm | 154 mm | 583 mm |
| Precipitation sum of the driest month | 0 mm | 57 mm | 151 mm |

## 6. Host Range

Knowledge of the host range of *F. circinatum* has been growing steadily, and recently reports of non-pine hosts have increased. For the purpose of this review, all known hosts of *F. circinatum* and their susceptibility ratings were compiled and assessed (Tables 3–6). The host list and susceptibility ratings were based on results of both field observations and experimental inoculations reported in peer-reviewed and "grey" literature, as well as from unpublished studies and the geo-database records compiled in this study. Such an extensive and integrated list has not previously been published because the information is scattered throughout numerous sources. We summarized these results and included data for 138 hosts (including 18 *Pinus* hybrids) tested in growth chamber, greenhouse, nursery and field inoculations or survey data from the wider environment.

Taxa from which data have been gathered include 96 species in the genus *Pinus* (including *Pinus* hybrids), 24 other tree species in 15 genera (*Abies*, *Cedrus*, *Chamaecyparis*, x*Cupressocyparis*, *Cupressus*, *Eucalyptus*, *Larix*, *Libocedrus*, *Picea*, *Podocarpus*, *Pseudotsuga*, *Sequoia*, *Sequoiadendron*, *Thuja,* and *Widdringtonia*) and 18 grass and herb species (see Tables 3–6; http://bit.do/phytoportal). In total, *F. circinatum* has been reported to infect 106 different plant species, including 67 *Pinus* species and 18 *Pinus* hybrids (Tables 3–5), as well as 6 non-pine tree species and 15 grass and herb species (Table 6). Overall, levels of susceptibility vary with the plant's age class, from recently emerged seedlings to mature trees. This variation is primarily due to the different behaviour of, and type of disease caused by, *F. circinatum* on plants of different ages. In pine seedlings, for example, *F. circinatum* essentially causes root disease (manifested as pre- and post-emergence, as well as late, damping-off), which is mainly seen in nursery situations, while the predominant symptom of infection in older or established pine trees are resinous cankers on the above-ground plant parts. Many species affected as seedlings in a nursery situation have not been seen to be affected as mature trees in a forest situation (Wingfield, unpublished). Because the behaviour and disease cycle of *F. circinatum* is likely to differ significantly in these two settings, the susceptibility ratings of seedlings and young plants (Table 3) were treated separately from those of older or mature trees (Table 4). Nevertheless, as nursery production is the primary route of *F. circinatum* transmission to the wider environment, a summary of the susceptibility ratings for both seedlings and mature trees is given below. This treatment allows for an assessment of the highest and lowest risk species that may serve as 'carriers' of the pathogen from the nursery to the forest and exhibit the disease in both settings.

*6.1. Host Susceptibility Ratings*

In this work, host susceptibility rating was based on the following categories: high, moderate-high, moderate, low-moderate, low, highly variable, unknown, and resistant. A host was considered resistant if no *F. circinatum* symptoms were detected after inoculation trials and natural infection with the fungus does not occur. The highly variable susceptibility category was assigned to the hosts (seedlings, young plants, young or mature trees) for which ranking varied in different studies from resistant to susceptible. Moreover, we did not include in these ratings endophytic infections, or asymptomatic plants (plants infected but seemingly healthy), as there is still very limited information about this particular lifestyle trait for *F. circinatum*.

The presence or absence of *F. circinatum* and severity of disease for 96 different *Pinus* taxa (including species, subspecies, varieties, and hybrids), either experimentally tested or observed, is reported (Tables 3–5). Unambiguous susceptibility rankings were obtained for 21 *Pinus* spp. (i.e., high susceptibility—three species; moderate susceptibility—four species; low susceptibility—13 species; resistant—one species) and 14 *Pinus* hybrids (i.e., moderate susceptibility—three hybrids; low susceptibility—11 hybrids). Nineteen *Pinus* species and seven *Pinus* hybrids were classified as having variable susceptibility, because different studies placed them in different susceptibility categories, while for 28 species the susceptibility classification was unknown. No symptoms of disease were observed on an additional 10 *Pinus* taxa (*P. cembra*, *P. contorta* var. *latifolia*, *P. heldreichii*, *P. mugo* subsp. *mugo*, *P. mugo* subsp. *rotundata*, *P. nigra* subsp. *nigra*, *P. nigra* subsp. *pallasiana*, *P. peuce*, *P. sylvestris* var. *hamata*, and *P. wallichiana*) which were monitored and systematically inspected for *F. circinatum* in the field (http://bit.do/phytoportal). Because these trees were monitored in areas where the pathogen has not been reported, the susceptibility or resistance status remains unknown. It can therefore be concluded that 67 *Pinus* species and 18 *Pinus* hybrids are known to be susceptible to *F. circinatum* based on artificial inoculation and natural infection observations (see Tables 3–5).

Susceptibility ratings for *F. circinatum* were analysed separately for different age classes of *Pinus* and non-*Pinus* hosts; i.e., seedlings and young plants (recently emerged pine seedling and plants, ≤10 years old) or mature trees (≥11 years). When only seedlings and plants of *Pinus* were considered, a total of 18 species were rated as highly susceptible, four as moderate-highly susceptible, 17 as moderately susceptible, seven as moderately-low susceptible, 22 as low susceptible, and one as resistant (*P. koraiensis*) to *F. circinatum*. Twelve *Pinus* species known to be hosts of *F. circinatum* and nine of the species have unknown susceptibility at the seedling stage (Table 3). When mature trees were considered separately, only a single pine species (*P. radiata*) was rated as highly susceptible, five species were rated as having low-moderate susceptibility, and four as having low susceptibility to *F. circinatum*. Twenty-six mature *Pinus* species known to be hosts of *F. circinatum* and 23 of species have unknown susceptibility rating to *F. circinatum* (Table 4). Three *Pinus* species in both age classes have highly variable susceptibility to *F. circinatum* (Tables 3 and 4 ᴛₐᵦₗₑ₃; ᴛₐᵦₗₑ₄).

Two pine species, *P. densiflora* and *P. koraiensis*, have been recorded as resistant to *F. circinatum* in 3–4 year old seedling inoculation trials conducted in greenhouses [59,135]. However, *F. circinatum* has been isolated from *P. densiflora* trees in Japan although the susceptibility of this host as a mature tree was not rated [136]. Therefore, we consider *P. densiflora* to have highly variable susceptibility to *F. circinatum* and the only truly resistant *Pinus* species to be *P. koraiensis*.

Non-pine tree species are generally only weakly susceptible or are resistant to *F. circinatum.* The susceptibility of non-pine hosts to *F. circinatum* was tested or observed on 24 tree and 18 herbaceous species (Table 6). Three conifer species (*Larix kaempferi*, *Libocedrus decurrens*, *Pseudotsuga menziesii*) were categorised as having low level of susceptibility. Another three conifer species (*Abies alba*, *Larix decidua*, *Picea abies*) were considered as having highly variable susceptibility because recently emerged seedlings were classed as susceptible to *F. circinatum*, whereas 2-year-old and older plants were considered resistant [8–10]. All other non-pine hosts, 18 non-pine tree species in 10 different genera, as well as three herbaceous species, were classed as resistant to *F. circinatum* (Table 6).

Although only three herbaceous plant species are classified as resistant to *F. circinatum* (Table 6), it must be noted that only a very limited number of herbaceous plants have been tested in this respect in pathogenicity assays. Fifteen species of herbaceous plants are known hosts of *F. circinatum* in the

wider environment, but their levels of susceptibility are unknown [5–7,26,137]. An additional consideration is that in some *P. radiata* plantations infected with *F. circinatum*, a number of herbaceous plants (Table 6) have been reported to be infected endophytically with *F. circinatum* [7]. It is thus clear that the full host range of *F. circinatum,* and susceptibility of each species, has yet to be elucidated.

**Table 3.** Susceptibility list of *Pinus* species seedlings and young plants (recently emerged pine seedling and plants, ≤10 years old) to *Fusarium circinatum.*

| Susceptibility/ Host Species[1] | Common English Names | Subgenus; Section; Subsection[2] | Type of Infection | Growth or Test Conditions | Seedlings and/or Plant Age[3] | References |
|---|---|---|---|---|---|---|
| | | | **Susceptibility high** | | | |
| *Pinus brutia* Ten. | Turkish pine, Calabrian pine, East Mediteranean pine, Brutia pine | Pinus; Pinus; Halepenses | Artificial | Growth chamber | 2 years | J. Martín-García, unpublished |
| *Pinus cembroides* Zucc. | Pinyon pine, Mexican nut pine | Strobus; Parrya; Cembroides | Artificial | Greenhouse | Unknown | [138] |
| *Pinus douglasiana* Martínez | Gordon's pine, Douglas pine | Pinus; Pinus; Ponderosae; 'Pseudostrobus Group' | Artificial | Greenhouse | Unknown | [138] |
| *Pinus greggii* Engelm. ex Parl. | Gregg's pine | Pinus; Pinus; Oocarpae; 'Oocarpa Group' | Artificial | Field | 2 years | [76] |
| *Pinus halepensis* Mill. | Aleppo pine | Pinus; Pinus; Halepenses | Artificial | Growth chamber | 2 years | J. Martín-García, unpublished |
| *Pinus hartwegii* Lindl. | Endlicher pine | Pinus; Pinus; Ponderosae; 'Montezumae Group' | Artificial | Greenhouse | Unknown | [138] |
| *Pinus herrerae* Martínez | Herrera's pine | Pinus; Pinus; Oocarpae; 'Teocote Group' | Artificial | Greenhouse | 12 weeks | [139] |
| *Pinus montezumae* Lamb. | Montezuma pine | Pinus; Pinus; Ponderosae; 'Montezumae Group' | Artificial | Greenhouse | Unknown | [138] |
| *Pinus mugo* Turra | Mountain pine, dwarf mountain pine | Pinus; Pinus, Pinus | Artificial | Growth chamber | Recently emerged | [9] |
| *Pinus mugo* Turra subsp. *uncinata* (Ramond ex DC.) Domin. | Swiss mountain pine | Pinus; Pinus, Pinus | Artificial | Growth chamber | Recently emerged | [8] |
| *Pinus nigra* J.F.Arnold | Austrian pine, black pine | Pinus; Pinus; Pinus | Artificial | Growth chamber | Recently emerged, 2 years | [8], J. Martín-García, unpublished |
| *Pinus patula* Schiede ex Schltdl. & Cham. | Patula pine, Jelecote pine, Mexican weeping pine, spreading-leaved pine | Pinus; Pinus; Oocarpae; 'Oocarpa Group' | Artificial | Greenhouse | 7–10 m | [71,75,140] |
| *Pinus pinaster* Aiton | Maritime pine | Pinus; Pinus; Pinus | Artificial | Growth chamber | Recently emerged, 2 years | [8,141], J. Martín-García, unpublished |
| *Pinus pseudostrobus* Lindl. | Smooth-bark Mexican pine | Pinus; Pinus; Ponderosae; 'Pseudostrobus Group' | Artificial | Greenhouse | Unknown | [138] |

| | | | | | | |
|---|---|---|---|---|---|---|
| *Pinus radiata* D. Don | Monterey pine, radiata pine, insignis pine | Pinus; Pinus; Attenuatae | Artificial | Growth chamber, greenhouse, field | Recently emerged, 3 months, 1 year, 2 years, 2-3 years, 3-4 years, unknown | [8,76,80,88,139,142,143] |
| *Pinus strobus* L. | Eastern white pine, northern white pine, white pine, Weymouth pine (British), and soft pine | Strobus; Strobus;Strobi | Artificial | Growth chamber | Recently emerged | [8] |
| *Pinus sylvestris* L. | Scots/Scotch pine | Pinus; Pinus; Pinus | Artificial | Growth chamber, greenhouse | Recently emerged, 1.5 y, 2 years | [8,10] J. Martín-García, unpublished |
| *Pinus taeda* L. | Loblolly pine | Pinus; Pinus; Australes | Artificial | Greenhouse | 1 year | [88] |
| **Susceptibility moderate-high** | | | | | | |
| *Pinus leiophylla* Schiede ex Schltdl. & Cham. | Chihuahua pine, smooth-leaf pine, yellow pine | Pinus; Pinus; Leiophyllae | Artificial | Field | 5–8 years | [144] |
| *Pinus patula* Schiede ex Schltdl. & Cham. | Patula pine, Jelecote pine, Mexican weeping pine, spreading-leaved pine | Pinus; Pinus; Oocarpae; 'Oocarpa Group' | Artificial | Greenhouse, Field | 3 months, 21 weeks, 2 years | [76,145,146] |
| *Pinus pinaster* Aiton | Maritime pine, cluster pine | Pinus; Pinus; Pinus | Artificial | Greenhouse | 6 months, 3 years | [141,147] |
| *Pinus taeda* L. | Loblolly pine | Pinus; Pinus; Australes | Artificial | Greenhouse, Field | 21 weeks, 4 years | [88,147] |
| **Susceptibility moderate** | | | | | | |
| *Pinus banksiana* Lamb. | Jack pine, scrub pine | Pinus; Pinus; Contortae | Artificial | Greenhouse | 3.5 years | [148] |
| *Pinus devoniana* Lindl. | Michoacán pine | Pinus; Pinus; Ponderosae 'Montezumae Group' | Artificial | Greenhouse, field | 12 weeks | [97,139] |
| *Pinus echinata* Mill. | Shortleaf pine | Pinus; Pinus; Australes | Artificial | Greenhouse | 3–4 years | [135] |
| *Pinus elliottii* Engelm. | Slash pine | Pinus; Pinus; Australes | Artificial | Greenhouse, field | 7–9 months, 8 years | [74,75,144] |
| *Pinus greggii* Engelm. ex Parl. | Gregg's pine | Pinus; Pinus; Oocarpae; 'Oocarpa Group' | Artificial | Greenhouse | 7 months | [75] |
| *Pinus halepensis* Mill. | Aleppo pine | Pinus; Pinus; Halepenses | Artificial | Greenhouse | 2 years, 3–4 years, unknown | [138,142,143] |
| *Pinus kesiya* Royle ex Gordon | Khasia pine, Khasi pine, Benguet pine | Pinus; Pinus; Pinus | Artificial | Field | 2 years | [76] |
| *Pinus leiophylla* Schiede ex Schltdl. & Cham. | Chihuahua pine, smooth-leaf pine, yellow pine | Pinus; Pinus; Leiophyllae | Artificial | Greenhouse | Unknown | [138] |
| *Pinus mugo* Turra subsp. *uncinata* (Ramond ex DC.) Domin. | Swiss mountain pine | Pinus; Pinus, Pinus | Artificial | Greenhouse | 2 years | [148] |

| | | | | | | |
|---|---|---|---|---|---|---|
| *Pinus nigra* J.F.Arnold | Austrian pine, black pine | Pinus; Pinus; Pinus | Artificial | Field | Unknown | [8] |
| *Pinus palustris* Mill. | Longleaf pine | Pinus; Pinus; Australes | Natural | Nursery | Recently emerged | [149] |
| *Pinus pinaster* Aiton | Maritime pine, cluster pine | Pinus; Pinus; Pinus | Artificial | Greenhouse, field | 2 years, Unknown | [8,143] |
| *Pinus pringlei* Shaw | Pringle´s pine | Pinus; Pinus; Oocarpae; 'Oocarpa Group' | Artificial | Greenhouse | Unknown | [138] |
| *Pinus strobus* L. | Eastern white pine, northern white pine, white pine, Weymouth pine (British), and soft pine | Strobus; Strobus;Strobi | Artificial | Greenhouse | 3.5 years | [150] |
| *Pinus sylvestris* L. | Scots/Scotch pine | Pinus; Pinus; Pinus | Artificial | Greenhouse | 2y, 3.5 years | [143,150] |
| *Pinus tecunumanii* F.Schwerdtf. ex Eguiluz & J.P.Perry | Schwerdtfeger's Pine, Tecun Uman Pine | Pinus; Pinus; Oocarpae; 'Oocarpa Group' | Artificial | Greenhouse | 3 months, 12 weeks, 6–8 months (high elevation origin) | [139,145,146] |
| *Pinus virginiana* Mill. | Virginia pine, Jersey pine, scrub pine | Pinus; Pinus; Contortae | Artificial | Greenhouse | 3–4 years | [135] |
| **Susceptibility low-moderate** | | | | | | |
| *Pinus caribaea* Morelet | Caribbean pine | Pinus; Pinus; Australes | Artificial | Greenhouse | 7 months | [75] |
| *Pinus elliottii* Engelm. | Slash pine | Pinus; Pinus; Australes | Artificial | Field | 2 years | [76] |
| *Pinus mugo* Turra subsp. *uncinata* (Ramond ex DC.) Domin. | Swiss mountain pine | Pinus; Pinus, Pinus | Artificial | Field | Unknown | [8] |
| *Pinus pinaster* Aiton | Maritime pine, cluster pine | Pinus; Pinus; Pinus | Artificial | Greenhouse | 2 years | [148] |
| *Pinus sylvestris* L. | Scots/Scotch pine | Pinus; Pinus; Pinus | Artificial | Growth chamber, greenhouse | Recently emerged, 2 years | [9,148] |
| *Pinus taeda* L. | Loblolly pine | Pinus; Pinus; Australes | Artificial | Greenhouse | 7–9 months, 1 year | [74,151] |
| *Pinus thunbergii* Parl. | black pine, Japanese black pine, Japanese pine | Pinus; Pinus; Pinus | Artificial | Greenhouse | 3–4 years | [135] |
| **Susceptibility low** | | | | | | |
| *Pinus ayacahuite* Ehrenb. ex Schltdl. | Mexican white pine, ayacahuite pine | Strobus; Strobus; Strobi | Artificial | Greenhouse | Unknown | [138] |
| *Pinus canariensis* C.Sm. | Canary Island pine | Pinus; Pinus; Canarienses | Artificial | Greenhouse | 2–3 years, 3–4 years | [3,142] |
| *Pinus caribaea* Morelet | Caribbean pine | Pinus; Pinus; Australes | Artificial | Greenhouse | 3 months | [139] |
| *Pinus clausa* (Chapm. ex Engelm.) Vasey ex Sarg. | Sand pine, Florida spruce pine, Alabama pine | Pinus; Pinus; Contortae | Artificial, natural | Greenhouse, field | 18 months, 6–8 years | [82] |
| *Pinus glabra* Walter | Spruce pine | Pinus; Pinus; Australes | Artificial, natural | Greenhouse, field | 5 months, unknown | [152] |
| *Pinus jaliscana* Perez de la Rosa | Jalisco pine | Pinus; Pinus; Oocarpae; 'Oocarpa Group' | Artificial | Greenhouse | 3 months | [139] |

| Species | Common name | Classification | Inoculation | Location | Age | Reference |
|---|---|---|---|---|---|---|
| *Pinus luchuensis* Mayr | Luchu/Ryukyu pine | Pinus; Pinus; Pinus | Artificial | Greenhouse | 2 years | [136] |
| *Pinus maximinoi* H.E. Moore | Thinleaf pine | Pinus; Pinus; Ponderosae; 'Pseudostrobus Group' | Artificial | Greenhouse | 3 months, 7 months, 8–9 months | [56,74,145] |
| *Pinus monophylla* Torr. & Frém. | Singleleaf pinyon pine | Strobus; Parrya; Cembroides | Natural | Field | Unknown | [79] |
| *Pinus nigra* J.F.Arnold | Austrian pine, black pine | Pinus; Pinus; Pinus | Artificial | Greenhouse | 2 years, 3.5 years | [143,150,153] |
| *Pinus occidentalis Sw.* | Western Indian pine | Pinus; Pinus; Australes | Natural | Field | Unknown | [53] |
| *Pinus oocarpa Schiede ex Schltdl.* | Ocote pine, Egg-cone pine, hazelnut pine | Pinus; Pinus; Oocarpae; 'Oocarpa Group' | Artificial | Greenhouse | 3 months, 7 months | [75,139,145] |
| *Pinus pringlei Shaw* | Pringle's pine | Pinus; Pinus; Oocarpae; 'Oocarpa Group' | Artificial | Greenhouse | 7 months | [75] |
| *Pinus pseudostrobus Lindl. syn. Pinus oaxacana* Mirov | Smooth-bark Mexican pine | Pinus; Pinus; Ponderosae; 'Pseudostrobus Group' | Natural, artificial | Field, greenhouse | Unknown, 7 months | [74,97] |
| *Pinus pungens* Lamb. | Table mountain pine | Pinus; Pinus; Australes | Artificial | Unknown | Unknown | [53] |
| *Pinus resinosa* Aiton | Red pine, Norway pine | Pinus; Pinus; Pinus | Artificial | Greenhouse | 3.5 years | [150] |
| *Pinus rigida* Mill. | Pitch pine | Pinus; Pinus; Australes | Artificial | Greenhouse | 3–4 years | [135] |
| *Pinus serotina* Michx. | Pond pine, marsh pine, pocosin pine | Pinus; Pinus; Australes | Artificial | Greenhouse | 1 year | [53,151] |
| *Pinus sylvestris* L. | Scots/Scotch pine | Pinus; Pinus; Pinus | Artificial | Field | 2 years, Unknown | [8] |
| *Pinus taeda* L. | Loblolly pine | Pinus; Pinus; Australes | Artificial | Field | 8 years | [140] |
| *Pinus tecunumanii* F.Schwerdtf. ex Eguiluz & J.P.Perry | Schwerdtfeger's Pine, Tecun Uman Pine | Pinus; Pinus; Oocarpae; 'Oocarpa Group' | Artificial | Greenhouse, field | 3 months, 12 weeks, 6–8 months, 8 years (low-elevation origin) | [56,74,139,144–146] |
| *Pinus thunbergii* Parl. | Black pine, Japanese black pine, Japanese pine | Pinus; Pinus; Pinus | Artificial | Greenhouse | 3–4 years | [142] |
| **Resistant** | | | | | | |
| *Pinus koraiensis* Siebold and Zucc. | Korean pine | Strobus; Strobus; Cembrae | Artificial | Greenhouse | 3 years | [59] |
| **Highly variable susceptibility** | | | | | | |
| *Pinus densiflora* Siebold and Zucc. | Japanese red pine, Korean red pine | Pinus; Pinus; Pinus | Artificial | Greenhouse | 3–4 years | [59,135,136] |
| *Pinus muricata* D. Don | Bishop pine | Pinus; Pinus; Attenuatae; | Artificial | Greenhouse | Unknown | [79] |
| *Pinus pinea L.* | Italian stone pine, umbrella pine | Pinus; Pinus; Pineae | Artificial | Greenhouse, growth chambers | 6 months, 2 years, 3-4 years | [35,37,143,154], J. Martín-García, unpublished |
| **Susceptible, unknown susceptibility rate** | | | | | | |
| *Pinus attenuata* Lemmon | Knobcone pine, Narrowcone pine | Pinus; Pinus; Attenuatae | Artificial | Greenhouse | Unknown | [79,154] |
| *Pinus brutia* Ten. var. | Eldarica pine | Pinus; Pinus; Halepenses | Artificial | Greenhouse | Unknown | [79] |

| | | | | | | |
|---|---|---|---|---|---|---|
| *eldarica* (Medw.) Silba | | | | | | |
| *Pinus canariensis* C.Sm. | Canary Island pine | Pinus; Pinus; Canarienses | Artificial | Greenhouse | Unknown | [79] |
| *Pinus contorta* Douglas ex Loudon | Shore pine, lodgepole pine | Pinus; Pinus; Contortae | Artificial | Greenhouse | Unknown | [141] |
| *Pinus coulteri* D. Don | Coulter pine, big-cone pine | Pinus; Pinus; Ponderosae; 'Sabinianae Group' | Artificial | Greenhouse | Unknown | [79,154] |
| *Pinus halepensis* Mill. | Aleppo pine | Pinus; Pinus; Halepenses | Artificial | Nursery | 2 years | [35] |
| *Pinus jeffreyi* A.Murray bis | Jeffrey pine | Pinus; Pinus; Ponderosae | Artificial | Greenhouse | Unknown | [79] |
| *Pinus lambertiana* Douglas | Sugar pine | Strobus; Strobus; Strobi | Artificial | Greenhouse | Unknown | [79] |
| *Pinus nigra* J.F. Arnold subsp. *laricio* (Poir.) Maire | Corsican pine | Pinus; Pinus; Pinus | Natural | Nursery | <6 months | A.V. Sanz-Ros, unpublished |
| *Pinus pinaster* Aiton | Maritime pine, cluster pine | Pinus; Pinus; Pinus | Artificial | Nursery | 2 years | [36] |
| *Pinus ponderosa* Douglas ex Loudon | Ponderosa pine, bull pine, blackjack pine | Pinus; Pinus; Ponderosae | Artificial | Greenhouse | Unknown | [79] |
| *Pinus sabiniana* Douglas ex D. Don | Gray/foothill/digger pine | Pinus; Pinus; Ponderosae; 'Sabinianae Group' | Artificial | Greenhouse | Unknown | [79] |

[1] Host taxonomy is based on Zanoni, Farjon [155]; [2] Subgenus; Section; Subsection is based on Price et al. [156]; [3] Seedlings and plants age: w—week, m—month, y—year.

**Table 4.** Susceptibility list of *Pinus* species mature trees (≥11 years) to *Fusarium circinatum.*

| Susceptibility/ Host Species[1] | Sampling site | Status of Host | Tree Age, Years | Reference |
|---|---|---|---|---|
| **Susceptibility high** | | | | |
| *Pinus radiata* D. Don | Plantation, unknown | Exotic, native | Unknown, 20y | [12,34,128,154,157] |
| **Susceptibility low-moderate** | | | | |
| *Pinus discolor* D.K. Bailey & Hawksw. | Unknown | Native | Unknown | [138] |
| *Pinus douglasiana* Martinéz | Unknown | Native | Unknown | [138] |
| *Pinus durangensis* Martinéz | Unknown | Native | Unknown | [138] |
| *Pinus halepensis* Mill. | Unknown | Exotic | Unknown | [138] |
| *Pinus leiophylla* Schiede ex Schltdl. & Cham. | Unknown | Native | Unknown | [138] |
| **Susceptibility low** | | | | |
| *Pinus ayacahuite* Ehrenb. ex Schltdl. | Unknown | Native | Unknown | [138] |
| *Pinus canariensis* C. Sm. | Urban trees | Exotic | Unknown | [154] |
| *Pinus luchuensis* Mayr | Greenhouse | Native | 11–19 years | [136] |
| *Pinus pinaster* Aiton | Plantation | Native | Unknown | |
| **Highly variable susceptibility** | | | | |

| | | | | |
|---|---|---|---|---|
| *Pinus densiflora* Siebold and Zucc. | Unknown | Native | Unknown | [136] |
| *Pinus muricata* D. Don | Unknown, Plantation | Native | Unknown, 12-13 years | [80,153] |
| *Pinus pinea* L. | Plantation, urban trees | Exotic, native | Unknown | [35,154] |
| **Susceptible, unknown susceptibility rate** | | | | |
| *Pinus arizonica* Engelm. | Plantation, natural forest | Native | Unknown | [43,97,138] |
| *Pinus armandii* Franch. | Natural forest | Native | Unknown | [136] |
| *Pinus attenuata* Lemm | Unknown | Native | Unknown | [154,158] |
| *Pinus canariensis* C. Sm. | Unknown | Exotic | Unknown | [15,80] |
| *Pinus cembroides* Zucc. | Unknown | Native | Unknown | [138] |
| *Pinus contorta* Douglas ex Loudon | Unknown | Native | Unknown | [154] |
| *Pinus contorta* Douglas ex Loudon var. *contorta* | Natural forest | Native | Unknown | [158] |
| *Pinus coulteri* D. Don | Unknown | Native | Unknown | [154] |
| *Pinus elliottii* Engelm. var. *densa* (Little and Dorman) E. Murray | Plantation | Native | Unknown | [17] |
| *Pinus elliottii* Engelm. var. *elliottii* | Plantation, natural forest | Native, exotic | Unknown | [13,159] |
| *Pinus greggii* Engelm. ex Parl. | Unknown | Exotic | Unknown | [74,76] |
| *Pinus halepensis* Mill. | Unknown, urban trees | Exotic, native | Unknown | [35,80] |
| *Pinus hartwegii* Lindl. | Plantation and natural forest | Native | Unknown | [40,138] |
| *Pinus kesiya* Royle. ex Gordon | Plantation | Exotic | Unknown | [56] |
| *Pinus leiophylla* Schiede ex Schltdl. & Cham. var. *leiophylla* | Natural forest | Native | Unknown | [120] |
| *Pinus montezumae* Lamb. | Unknown | Native | Unknown | [138] |
| *Pinus patula* Schiede ex Schltdl. & Cham. | Plantation | Exotic | Unknown | [56] |
| *Pinus ponderosa* Douglas ex Lawson | Plantation | Native | Unknown | [154] |
| *Pinus pringlei* Shaw | Unknown | Native | Unknown | [138] |
| *Pinus pseudostrobus* Lindl. | Unknown | Native | Unknown | [138] |
| *Pinus pseudostrobus* Lindl. var. *apulcensis* (Lindl.) Shaw | Plantation and natural forest | Native | Unknown | [40,97,138] |
| *Pinus radiata x attenuata* | Natural forest | Native | Unknown | [158] |
| *Pinus sabiniana* Douglas ex D. Don | Unknown | Native | Unknown | [154,158] |
| *Pinus taeda* L. | Unknown, greenhouse | Native | Unknown, 21 years | [151,160] |
| *Pinus thunbergii* Parl. | Roadside, golf course | Native | Unknown | [161] |

| *Pinus torreyana* Parry ex Carrière | Unknown | Native | Unknown | [154,158] |

[1] Host taxonomy is based on Zanoni, Farjon [155].

*6.2. Species with Variable Susceptibility Ratings*

Susceptibility ratings were variable for 19 *Pinus* species and seven *Pinus* hybrids due to the fact that different studies classified them in different susceptibility categories (see Tables 3-5). Variable susceptibility ranging from susceptible to resistant was found in two mature pine hosts: *P. muricata* and *P. pinea*. The highest discrepancy between susceptible ratings was noted for species classified as having both high and low susceptibility: *P. nigra*, *P. pseudostrobus*, *P. sylvestris*, and *P. taeda*. Two *Pinus* species (*P. pinaster* and *P. mugo* subsp. *uncinata*) were ranked as having both high and low-moderate susceptibility. Slight differences in susceptibility ratings, i.e., those in adjacent ranking categories, were recorded for 11 pine hosts: *P. canariensis*, *P. caribaea*, *P. elliottii*, *P. greggii*, *P. halepensis*, *P. leiophylla*, *P. patula*, *P. pringlei*, *P. strobus*, *P. tecunumanii,* and *P. thunbergii* (Tables 3 and 4 Table 3; Table 4). These minor discrepancies in susceptibility ratings are most likely due to experimental variation and interpretation of the categories by various authors and are not discussed further in this review, while the species with greater discrepancies in susceptibility ratings are discussed in more detail below.

Among the seven *Pinus* hybrids, we found ambiguous classifications for those with *P. tecunumanii* and *P. greggii* as one of the parental species (Table 5). Seeds for these two *Pinus* species have been sourced from different locations in Mexico, and susceptibility to *F. circinatum* varies based on seed sources [139]. *Pinus tecunumanii* originated from high elevations, and *P. greggii* from northern Mexico (i.e., *P. greggii* var. *greggi*), are respectively more susceptible to *F. circinatum* than low-elevation *P. tecunumanii* and *P. greggii* from southern Mexico (i.e., *P. greggii* var. *australis*) [75,76]. Failure to mention such provenance information has accordingly resulted in apparently inconsistent pathogenicity data in some published studies.

Mature *P. pinea* was rated as resistant to *F. circinatum* by Gordon et al. [154], but in urban conditions in Italy the host was considered susceptible [34]. Furthermore, seedlings of *P. pinea* were rated as having low susceptibility [37,142,143,148] and unknown susceptibility [35], but *P. pinea* seedlings of Spanish origin (6 months, 12–14 cm) were very susceptible and taller seedlings (6 months, > 16 cm) of the same origin were resistant (Martín-García, unpublished). Therefore, we consider that *P. pinea* has highly variable susceptibility. The results of *P. muricata* inoculation tests by Schmale & Gordon [153] indicated a wide range of variation in susceptibility: 27% of *P. muricata* trees were considered resistant, while others were susceptible to *F. circinatum*. In other studies, *P. muricata* seedlings and mature trees were shown to be susceptible to *F. circinatum* [79,80]. Thus, we considered *P. muricata* to be highly variable in susceptibility to *F. circinatum*.

*Pinus sylvestris* seedlings, both recently emerged and 1.5-year-old seedlings, of Spanish, Czech, and Scottish origin, were classified as highly susceptible in growth chamber and greenhouse experiments by Martinez-Alvarez et al. [8], Martín-García et al. [10], and Woodward, unpublished. In contrast, low susceptibility of *P. sylvestris* seedlings of the Spanish origin was observed in a field trial [8]. No data are available for the susceptibility of mature *P. sylvestris* trees.

*Pinus nigra* seedlings of different provenances were highly susceptible to *F. circinatum* in growth chamber experiments [8, Martín-García, unpublished]. In contrast, in greenhouse experiments 2- and 3.5-years old seedlings of *P. nigra* were found to have low levels of susceptibility [143,150]. This discrepancy could be a result of different environmental conditions (temperature, humidity) in the growth chamber and greenhouse as compared to field conditions. These kinds of experiments should be done under the same environmental conditions. The variation in susceptibility of 2-year-old *P. nigra* seedlings of Spanish origin was quite high between individuals, indicating that some individuals in this species may be at higher risk of damage from *F. circinatum* [143]. Less than six-months-old seedlings of *P. nigra* subsp. *laricio* of the Spanish origin were found naturally (not artificially inoculated) infected in nurseries (Sanz-Ros, unpublished), and this was considered as unknown susceptibility. No data are available for the susceptibility of mature *P. nigra*.

One-year-old *P. taeda* seedlings in South Africa were found to be susceptible to *F. circinatum* [88], in contrast to other studies showing that *P. taeda* was highly tolerant to *F. circinatum* in South Africa [76,139]. Seven-month-old seedlings of *P. pseudostrobus* were ranked as having low susceptibility and were more tolerant to the pathogen than *P. taeda* [71]. However, mature trees of both *P. taeda* and *P. pseudostrobus* were found to be susceptible to *F. circinatum* [138,151]. These contradictory results could relate to the age of trees being inoculated and differences between natural infections on established trees and those arising from artificial inoculations.

For *P. pinaster*, recently emerged and 3-year-old seedlings were deemed to be highly susceptible in growth chamber and greenhouse experiments by Martinez-Alvarez et al. [8] and Vivas et al. [141]. However, Iturritxa et al. [148] considered 2-year-old *P. pinaster* seedlings of the Spanish origin to have low-moderate susceptibility in greenhouse experiments. In addition, variation in resistance was found between Spanish provenances of *P. pinaster*, e.g., one provenance was estimated to be more resistant than the three other tested provenances [148].

Recently emerged seedlings of *P. mugo* and *P. mugo* subsp. *uncinata* were classed as highly susceptible in growth chamber experiments [8,9], whereas seedlings of the same provenance of *P. mugo* subsp. *uncinata* in the field were classified as having low-moderate susceptibility [8]. This variation could be purely due to differences in field and growth chamber experiments, yet other studies have suggested that the levels of susceptibility of *Pinus* spp. demonstrated in greenhouse inoculations to correlate quite well with incidence of the disease observed in the field [142].

The examples given above demonstrate the wide variation in rankings of particular species to infection by *F. circinatum*. There are several possible explanations for the variation in susceptibility ratings between reports which are centred on different aspects of the disease triangle. These differences include between-provenance variation in susceptibility, an interaction between environmental conditions and relative host susceptibility, variation in the relative virulence of *F. circinatum* haplotypes or populations on different host species, different inoculum densities, variation in the relative susceptibility of a species at different ages, differences in the interpretation of susceptibility categories by different assessors, and the comparison of different numbers of host species by different authors. Some of these aspects are discussed below.

The level of resistance to infection by *F. circinatum* has been found to vary between *Pinus* species and among provenances, possibly a function of long-term exposure to the pathogen. For example, most *Pinus* species that occur in Central America and the Southeastern USA, the proposed centre of *F. circinatum* evolution, or where it has been present for many years, display a high level of resistance [139]. However, *P. patula* as well as some other *Pinus* species that occur naturally in the putative natural range of the pathogen in Mexico and would be assumed to have some level of resistance to pitch cankerare highly susceptible. As another example, in South Carolina, 21-year-old clonal *P. taeda* trees in a seed orchard had a wide range of susceptibility to *F. circinatum* [151]. These apparent discrepancies have been suggested to point towards other factors that may be involved in susceptibility [120,139]. In addition to the life history traits of the plant host, these also include environmental factors such as insect feeding preferences or site conditions.

Variation in the susceptibility of a host, particularly if a host is ranked in different adjacent susceptibility categories, could be due to experimental variation, especially if different studies used different numbers of trees, inoculation procedures, or challenging conditions, e.g., temperature or irrigation. Alternatively, differences could simply be due to differences in the interpretation of susceptibility categories. An extension of these author-based sources of variation is the lack of information about the host provenance, similar to the problems associated with the susceptibility data for certain *Pinus* hybrids, mentioned earlier. For example, there are two varieties of *P. elliottii* in the Southeastern USA, *P. elliottii* var. *densa*, distributed from central to southern Florida, and *P. elliottii* var. *elliottii*, distributed from central Florida to South Carolina and westward to Louisiana. Although significant differences in the susceptibility to *F. circinatum* between these varieties have been observed [17], generally both varieties are referred to as *P. elliottii*, and most studies do not distinguish the variety used when reporting the effects of PPC. This oversight could lead to perceived variation in susceptibility rankings between studies, where varieties or provenances are not always recorded.

Additional sources of variation are the amounts of inoculum used in artificial inoculation trials or the method of inoculation (e.g., under-bark mycelium insertion, under-bark spore suspension insertion, soil drenching).

A wide range of factors affect host susceptibility to *F. circinatum*, as well as assessment and interpretation of pathogenicity data. Susceptibility ranking depends primarily on host species and host variety or provenance with the associated specific genetic variation within these hosts and virulence among *F. circinatum* isolates. However, susceptibility is also heavily influenced by environmental factors, many of which are understudied and poorly understood. For instance, the expression of PPC disease was influenced by maternal effects of *P. pinaster*, given that environmental conditions experienced by mother trees affected tolerance in offspring [162]. Another important confounding issue relates to the multiplicity of methodologies employed to rate susceptibility to the PPC pathogen. Plant age also plays an important role in the susceptibility of trees to *F. circinatum*. It is known that seedlings have different mechanisms of resistance to infection by pathogens when compared to mature plants, and there is a need to equate knowledge from natural infections in the forest and wider environment to those based on seedling or nursery problems. For this reason, the susceptibility of seedlings and young plants (Table 3) (corresponding to the disease in nurseries) and mature trees (Table 4) (corresponding to the disease in wider environment) are treated separately in this review. Furthermore, much work needs to be done to fully understand the interactions between the host, the environment, and the pathogen and their influences on susceptibility.

**Table 5.** Susceptibility list of *Pinus* hybrids to *Fusarium circinatum*.

| Susceptibility/ Host Species[1] | Sampling Site | Type of Infection | Plants Stage Tested | Reference |
|---|---|---|---|---|
| **Susceptibility moderate-high** | | | | |
| *Pinus greggii* x *Pinus tecunumanii* | Field | Artificial | Established plantation trees | [76] |
| *Pinus patula* x *Pinus greggii* | Field | Artificial | Established plantation trees | [76] |
| *Pinus patula* x *Pinus greggii var. australis* (from southern Mexico) | Greenhouse | Artificial | Nursery plants established from rooted cuttings | [75] |
| *Pinus patula* x *Pinus radiata* | Field | Artificial | Established plantation trees | [76] |
| **Susceptibility moderate** | | | | |
| *Pinus patula* x *Pinus greggii* var. *greggii* | Nursery | Natural | Nursery plants established from rooted cuttings | [163] |
| *Pinus patula* x *Pinus pringlei* | Greenhouse | Artificial | Nursery plants established from rooted cuttings | [75] |
| *Pinus patula* x *Pinus tecunumanii* (High elevation source) | Greenhouse | Artificial | Nursery plants established from rooted cuttings | [75] |
| *Pinus patula* x *Pinus tecunumanii* | Field | Artificial | Established plantation trees | [76] |
| **Susceptibility low-moderate** | | | | |
| *Pinus patula* x *Pinus elliottii* | Greenhouse | Artificial | Nursery plants established from rooted cuttings | [75] |
| *Pinus patula* x *Pinus tecunumanii* (High elevation source) | Nursery | Natural | Nursery plants established from rooted cuttings | [163] |
| *Pinus rigida* x *Pinus taeda* | Greenhouse | Artificial | Nursery plants established from rooted cuttings | [135] |
| **Susceptibility low** | | | | |
| *Pinus caribaea* x *Pinus oocarpa* | Greenhouse | Artificial | Nursery plants established from rooted cuttings | [75] |
| *Pinus elliottii* x *Pinus caribaea* | Field, greenhouse | Artificial | Established plantation trees, nursery plants established from rooted cuttings | [75,76] |
| *Pinus elliottii* x *Pinus taeda* | Greenhouse | Artificial | Nursery plants established from rooted cuttings | [75] |
| *Pinus elliottii* x *Pinus tecunumanii* | Greenhouse | Artificial | Nursery plants established from rooted cuttings | [75] |
| *Pinus patula* x *Pinus caribaea* var. *hondurensis* | Nursery | Natural | Nursery plants established from rooted cuttings | [163] |
| *Pinus patula* x *Pinus greggii* var. *australis* | Nursery | Natural | Nursery plants established from rooted cuttings | [163] |

| *Pinus patula* x *Pinus oocarpa* | Field, greenhouse | Artificial | Established plantation trees, nursery plants established from rooted cuttings | [75,76] |
|---|---|---|---|---|
| *Pinus patula* x *Pinus tecunumanii* | Field | Artificial | Established plantation trees | [76] |
| *Pinus patula* x *Pinus tecunumanii* (Low elevation source) | Greenhouse, nursery | Artificial, natural | Nursery plants established from rooted cuttings | [75,163] |
| *Pinus taeda* x *Pinus tecunumanii* | Field | Artificial | Established plantation trees | [76] |
| *Pinus tecunumanii* (High elevation source) x *Pinus caribaea* | Greenhouse | Artificial | Nursery plants established from rooted cuttings | [75] |
| *Pinus tecunumanii* (High elevation source) x *Pinus oocarpa* | Greenhouse | Artificial | Nursery plants established from rooted cuttings | [75] |
| *Pinus tecunumanii* (Low elevation source) x *Pinus caribaea* | Greenhouse | Artificial | Nursery plants established from rooted cuttings | [75,164] |

[1] For *Pinus tecunumanii* and *Pinus greggii*, seed sources are indicated where known.

**Table 6.** Susceptibility list of non-pine species to *Fusarium circinatum*.

| Susceptibility/Host Species [1] | Common English Names | Type of Infection | Growth or Test Conditions | Host Age, Years | References |
|---|---|---|---|---|---|
| **Susceptibility low** | | | | | |
| *Larix kaempferi* (Lamb.) Carr. | Japanese larch | Artificial | Greenhouse | unknown | [3] |
| *Libocedrus decurrens* (Torr.) Florin | Incense cedar | Artificial | Greenhouse | unknown | [8] |
| *Pseudotsuga menziesii* (Mirb.) Franco | Douglas-fir | Artificial, natural | Greenhouse, field | unknown | [3] |
| **Resistant** | | | | | |
| *Cedrus atlantica* (Endl.) G.Manetti ex Carrière | Atlas cedar | Artificial | Greenhouse | Unknown | [8] |
| *Chamaecyparis lawsoniana* (A. Murray) Parl. | Port Orford cedar, Lawson cypress | Artificial | Greenhouse | Unknown | [8] |
| *Cupressocyparis × leylandii* (A.B.Jacks. & Dallim.) Dallim. | Leyland cypress | Artificial | Greenhouse | Unknown | [8] |
| *Cupressus macrocarpa* Hartw. ex Gordon | Monterey cypress | Artificial | Greenhouse | Recently emerged | [165] |
| *Eucalyptus regnans* F. Muell. | Mountain ash, swamp gum | Artificial | Greenhouse | Recently emerged | [165] |
| *Eucalyptus globulus* Labill. | Australian fever tree, southern blue gum | Artificial | Greenhouse | Recently emerged | [165] |
| *Picea abies* (L.) H. Karst. | Norway spruce | Artificial | Greenhouse | 2 years | [10] |
| *Picea jezoensis* Carrière | Dark-bark spruce | Artificial | Greenhouse | Unknown | [3] |
| *Picea sitchensis* (Bong.) Carrière | Sitka spruce | Artificial | Greenhouse | Unknown | [3] |
| *Podocarpus latifolia* (Thunb.) R.Br. ex Mirb. | Broad-leaved yellowwood, real yellowwood | Artificial | Greenhouse | Unknown | [140] |
| *Podocarpus elongatus* Aiton L´Hérit. ex Pers. | Breede River yellowwood | Artificial | Greenhouse | Unknown | [140] |
| *Podocarpus henkelii* Stapf ex Dallim. & A.B.Jacks. | Henkel's yellowwood | Artificial | Greenhouse | Unknown | [140] |
| *Sequoia sempervirens* (D. Don) Endl. | Coast redwood | Artificial | Greenhouse | Unknown | [3] |
| *Sequoiadendron giganteum* (Lindl.) J.Buchh. | Giant sequoia | Artificial | Greenhouse | Unknown | [3,8] |
| *Thuja plicata* Donn ex D. Don | Western Redcedar | Artificial | Greenhouse | Unknown | [8] |
| *Widdringtonia schwartzii* (Marloth) Mast. | Willowmore Cedar, Baviaans Cedar, Willowmore Cypress | Artificial | Greenhouse | Unknown | [140] |
| *Widdringtonia cederbergensis* (Marsh) | Clanwilliam cedar, Cape cedar | Artificial | Greenhouse | Unknown | [140] |
| *Widdringtonia nodiflora* (L.) Powrie | Mountain cedar, mountain cypress | Artificial | Greenhouse | Unknown | [140] |
| **Resistant herbs** | | | | | |
| *Gladiolus* L. | Gladius, a sword, gladioli | Artificial | Greenhouse | Corms | [88] |
| *Lolium perenne* L. | Perennial ryegrass | Artificial | Greenhouse | Recently emerged | [165] |
| *Trifolium repens* L. | White clover | Artificial | Greenhouse | Recently emerged | [165] |

| Highly variable susceptibility | | | | | |
|---|---|---|---|---|---|
| *Abies alba* Mill. | European silver fir | Artificial | Greenhouse | Recently emerged, unknown | [8] |
| *Larix decidua* Mill. | European larch | Artificial | Greenhouse | Recently emerged, 2 years | [8,10] |
| *Picea glauca* (Moench) Voss | White spruce | Artificial | Greenhouse | Recently emerged, unknown | [3,8,9] |
| Susceptible, unknown susceptibility rate | | | | | |
| *Agrostis capillaris* L. | Common bent, colonial bent, or browntop | Natural | Plantation of *P. radiata* | Unknown | [7] |
| *Briza maxima* L. | Big quaking grass, great quaking grass | Natural | Nursey | Unknown | [6] |
| *Centaurea debeauxii* Godr. & Gren. | Meadow knapweed | Natural | Plantation of *P. radiata* | Unknown | [7] |
| *Cymbidium* sp. Sw. | Boat orchids | Natural | Unknown | Unknown | [26] |
| *Ehrharta erecta* Lamb. var. *erecta* (Hochst.) Pilg. | Panic veldtgrass | Natural | Nursey | Unknown | [6] |
| *Festuca arundinacea* Schreb. | Tall fescue | Natural | Field | Unknown | [5] |
| *Holcus lanatus* L. | Common velvet grass | Natural | Field | Unknown | [5] |
| *Hypochaeris radicata* L. | Cat´s ear | Natural | Plantation of *P. radiata* | Unknown | [7] |
| *Musa acuminata* Colla | Wild banana | Natural | Unknown | Unknown | [26] |
| *Pentameris pallida* (Thunb.) Galley & H.P.Linder | Pussy tail grass | Natural | Nursey | Unknown | [6] |
| *Pseudarrhenatherum longifolium* (Thore) Rouy | Oat grass | Natural | Plantation of *P. radiata* | Unknown | [7] |
| *Rubus ulmifolius* Schott | Elmleaf blackberry | Natural | Plantation of *P. radiata* | Unknown | [7] |
| *Sonchus oleraceus* L. | Common sowthistle | Natural | Plantation of *P. radiata* | Unknown | [7] |
| *Zea mays* L. | Maize | Natural | Unknown | Unknown | [137] |
| *Teucrium scorodonia* L. | Woodland germander, wood sage | Natural | Plantation of *P. radiata* | Unknown | [7] |

[1] Host taxonomy is based on Farjon [138].

## 6.3. Analyses of Geo-Database Data and Host Range

Globally, 138 plant hosts (96 *Pinus*, 24 non-*Pinus* woody species, and 18 grass and herb species) were surveyed for susceptibility to *F. circinatum* using reports involving visual and molecular detection. In terms of geography, our study reports *F. circinatum* from 14 countries (including four European countries) and considers the pathogen absent in 28 countries. The interactive map (http://bit.do/phytoportal) presents 6297 observational records, which represent reports from 87 plant species and shows the status of *F. circinatum* in the 41 countries for which there are data. The interactive map does not include data for 16 herbaceous plant species and 35 pine hosts (including 18 *Pinus* hybrids) because the material was experimentally tested in growth chambers/greenhouses and precise locations are unknown, but records for these hosts are reported in Tables 3-5. Conversely the 10 *Pinus* species that were surveyed in the field, but in areas free of *F. circinatum* (see Section 6.1), are included in the interactive map. However, these species are not included in Tables 3-5 as the susceptibility ratings remain unknown. In Europe, 31 different hosts were inspected for *F. circinatum* and symptoms of PPC, amongst which 11 hosts were found to be susceptible to *F. circinatum* (see interactive map: http://bit.do/phytoportal).

## 7. Conclusions

The information collated in this review was derived from a wide range of sources and represents the most comprehensive documentation on the global range of *F. circinatum* made to date. An unprecedented level of information about PPC was compiled using results from new disease surveys initiated as part of COST Action FP1406 "Pine Pitch Canker Strategies for Management of *Gibberella circinata* in Green Houses and Forests" (PINESTRENGTH), representing field observations and laboratory tests of over 6297 samples collected from different hosts.

*Fusarium circinatum* has now been reported from 14 countries in the world (Brazil, Chile, Colombia, France, Haiti, Italy, Japan, South Korea, Mexico, Portugal, Spain, South Africa, Uruguay, USA). The fungus is considered absent in 28 countries (24 European countries, Australia, New Zealand, Turkey, Israel), and it has been officially eradicated from Italy and France. The pathogen is present only in nurseries in Brazil, Chile, and Uruguay and has not yet been found on mature trees in the wider environment. The inclusion of sample information from poorly studied areas has provided improved knowledge of the distribution of *F. circinatum* worldwide. All records are available in the geo-database (http://bit.do/phytoportal), which will be active in the future, and updated information regarding the pathogen will be included.

The current distribution of *F. circinatum* was summarized using a number of fine scale climatic and topological variables. Only data points from the open environment were analysed, excluding nursery records. The moisture dependent *F. circinatum* has been detected inland far from the coast, where the pathogen may survive and potentially cause disease from −14.2°C (mean temperature of the coldest month) to +36.5°C (mean temperature of the warmest month), indicating that *F. circinatum* may establish in a wider range of areas than previously thought. This finding suggests that *F. circinatum* could thrive and cause considerable damage in nurseries from high latitudes or areas generally not considered suitable for PPC. This possibility should be taken into account when assessing the risk of nurseries in these areas.

This review summarizes data for 138 host species tested in growth chamber experiments, greenhouse, nursery, or field inoculations; or survey data from the wider environment. The species from which data were gathered include 96 species in the genus *Pinus*, 24 other tree species in fifteen genera, and 18 grass and herb species. *Fusarium circinatum* has been reported to infect 106 different host species, including 67 *Pinus* species, 18 *Pinus* hybrids, 6 non-pine tree species, and 15 grass and herb species. Levels of susceptibility to *F. circinatum* were analysed separately for different age classes of pine and non-pine hosts, classified as either seedlings (recently emerged seedlings and plants) or mature trees, due to the different disease cycle of *F. circinatum* in nursery and wider environmental conditions. Only one *Pinus* species, *P. koraiensis*, is considered resistant to *F. circinatum* based on greenhouse inoculations. Tree age clearly plays an important role in susceptibility to *F. circinatum*, and much work needs to be done to fully understand the interaction between host, environment and pathogen and their interactive influences on susceptibility and severity of infection.

Thus, the results in this study clearly support the need for standard protocols to be applied in pathogenicity tests to compare data in different conditions (including laboratories) and increase knowledge on the biology and epidemiology of *F. circinatum*.

Knowledge of the centre of origin and source populations of *F. circinatum* can aid in sourcing resistant species or valuable genetic material. Combined with information regarding introduction pathways, it can also help prevent further introductions by focusing quarantine measures and monitoring efforts where they are most effective. Given the risks posed by the movement of the pathogen, there is an urgent need for routine surveillance of *Pinus* and other species known to be susceptible to *F. circinatum*, as well as research on the importance of soil, native insects, and asymptomatic nursery plants for the spread of the disease [166]. Without maintained levels of surveillance, it is highly probable the pathogen will continue spreading to new areas and extending its host range.

**Supplementary Materials:** The following are available online at www.mdpi.com/1999-4907/11/7/724/s1, supporting information to Geo-database, Table S1: Data fields of the international *Fusarium circinatum* (FC) geo-database (for more information see: http://bit.do/phytoportal), GIS and map analyses, and statistical analyses.

**Author Contributions.** All authors sent monitoring data of the pathogen for geo-database and interactive map. Writing: R.D., M.S.M., B.G, J.M.G., P.V. Review: all authors. Editing: S.W., K.A. (Katarína Adamčíková). All authors have read, corrected and agreed to the published version of the manuscript.

**Funding:** This study was financially supported by COST Action FP1406 (PINESTRENGTH), the Estonian Science Foundation grant PSG136, the Forestry Commission, United Kingdom, the Phytophthora Research Centre Reg. No. CZ.02.1.01/0.0/0.0/15_003/0000453, a project co-financed by the European Regional Development Fund. ANSES is supported by a grant managed by the French National Research Agency (ANR) as part of the "Investissements d'Avenir" programme (ANR-11-LABX-0002-01, Laboratory of ExcellenceARBRE). SW was partly supported by BBSRC Grant reference BB/L012251/1 "Promoting resilience of UK tree species to novel pests & pathogens: ecological & evolutionary solutions (PROTREE)" jointly funded by BBSRC, Defra, ESRC, the Forestry Commission, NERC and the Scottish Government, under the Tree Health and Plant Biosecurity Initiative. Annual surveys in Switzerland were financially supported by the Swiss Federal Office for the Environment FOEN.

**Acknowledgments:** Andrea Kunova and Cristina Pizzatti are acknowledged for the assistance in the sampling. Thanks are due to Dina Ribeiro and Helena Marques from ICNF-Portuguese Forest Authority for providing location coordinates. We thank three anonymous reviewers for valuable corrections and suggestions.

**Conflicts of Interest:** The authors declare no conflict of interest.

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
