# Peer review of "Global Geographic Distribution and Host Range of Fusarium circinatum, the Causal Agent of Pine Pitch Canker"

_forests, doi:10.3390/f11070724_

Round 1
Reviewer 1 Report
This review is a gargantuan effort to summarize up to date global distribution of pine pitch canker, an important disease of pines, and exotic in much of its current distribution. The Phytopathological Portal, GIS database is very impressive. The manuscript is well written and a pleasure to read. Only minor comments:
-Author affiliations could be revised. For example, 47 and 48 are the same except for email.
-In some tables, a few columns were crowded. For example, column 3 in Tables 1 and 3. These could be altered to allow more space for crowded columns
-Quality of Figure 1-2 could be improved so that the legends are more legible
Author Response
Please see attached pdf file.

Reviewer 2 Report
This review article summarizes the global distribution of Fusarium circinatum, cause of pine pitch canker, with an emphasis on Europe. Pathogen diversity, future climatic range predictions, and host ranges and susceptibilities are also reviewed extensively. The manuscript is based on over 6,000 observations which revealed the presence of F. circinatum in 14 countries on 106 susceptible host species. An associated map developed in ArcGIS Online is referenced numerous times throughout the text and was an excellent complement to the work. Negative observations and eradicated populations are noted along with active infestations or populations. The review cited 164 sources, including grey literature, and although some self citations are present, they are necessary, and other work is adequately represented.
The manuscript is well written, and other than some minor issues and suggestions listed below, I recommend accepting pending these minor revisions. Some are merely suggestions, but there were several run-on sentences that require commas and some other issues. There is a lack of consistency in the use of Oxford commas which should be resolved . . . Oxford commas are identified in several places where they are used. References should have links added where possible (https://doi.org/...). Specific comments and suggestions:
Line 128: Comma after “documented”
Line 141: Pluralize “age”
Line 142: “are” should be “is” (Knowledge . . . is essential)
Line 151: Does EPPO or EU need to be defined here?
Line 162-163: Work by Swett et al. (2018, Forest Pathology, doi: 10.1111/efp.12422) suggests infection does not necessarily require wounding, though development of disease generally does
Line 176: Comma after “pathogens”
Line 184: Consider a semicolon after “circinatum” because of commas within preceding list
Line 188: comma after “hosts”
Line 211: comma after “methods”
Line 222: You use an Oxford comma here but nowhere else thus far in the manuscript and should be consistent.
Line 224: Comma after Galicia
Line 248: add “the” before “first”
Line 253: Don’t start a sentence with an abbreviation, write out “Pine pitch canker” here
Line 255: Comma after “destroyed”
Line 258: Delete comma after [35]
Line 272: Comma after “therefore”
Line 274: Suggest replacing “taken into account” with “considered”
Line 276: Delete “the”
Line 277: Don’t start a sentence with an abbreviation, write out “Pine pitch canker” here
Lines 286-294: Numerous Oxford commas used here, but not in most of the rest of the document, and half of this paragraph appears to be gray text
Line 314: “gregii” should be “greggii”
Line 320: Delete comma after [69-71]
Line 327: Does this mean landscape plantings of pine and Douglas-fir? Nursery stock? This is a bit vague and could be clarified
Line 330: Comma after “regulations”
Line 331: Comma after “California”
Line 335: “Year of finding” should perhaps be replaced with “year found”
Table 1: Try to expand second column so “Environment” occurs on one line
Line 345: Start new sentence with “see interactive map”
Line 354: Delete comma after “fungus”
Line 365: Comma after “complex”
Line 372: Delete “that”
Line 379: “is” should be “are”
Line 398: Populations should be singular
Line 406: Don’t start a sentence with an abbreviation (VCGs)
Line 413: delete comma after VCGs
Lines 418-419: “where reproduction of the pathogen is asexual” should perhaps be moved to after “Japan,” and a comma should be added before “only”
Line 437: Comma after “nurseries”
Line 453: Delete comma after “Portugal”
Line 456: Add “of” after “origin”
Line 459: Consider changing “shed light on” to something like “elucidate”
Line 469: Fusarium is not abbreviated in other section headings and should be written out here
Line 473: Oxford comma used here but not in most of the document
Line 476: I had to look up occult precipitation, a brief definition may be helpful here.
Lines 488-491: Because commas are used within list items, separate primary items by semi-colons
Line 497: It is unclear what “respectively” refers to; in the previous sentence, suitable habitat is listed first followed by optimal habitat. The smaller number in this sentence is reported first, though.
Line 530: Oxford comma is used here
Line 546: Comma after [124] can be deleted
Line 552: Why is latter used here? Maybe delete that or rewrite this sentence entirely, it is unclear
Line 561: Oxford comma is used here
Line 575: Comma should come before “and”, not after
Table 2: “emperature” should be “temperature” in fourth row
Line 594: Comma should come before “and”, not after
Line 604: Is there supposed to be an “x” before “Cubressocyparis”?
Line 612: add a comma after “late”
Line 640: should “or subspecies” be added because some of the ten listed are different subspecies within the same species?
Line 646: “Infections” can be singular
Lines 654-655: I don’t know what this sentence is trying to say; it appears it may start with a sentence fragment, but I’m not really sure and it should be rewritten. Same comment for sentence in lines 658-660
Line 706: Can delete comma after “tecunumanii”
Line 709: Can delete “of”
Line 714: Maybe use “taller” instead of “higher”
Line 716: Abbreviate Pinus
Lines 727-728: Hyphenate 2- and 3.5-year-old seedings
Line 730: Change to “These kinds of experiments”
Lines 733-734: Hyphenate six-months-old
Line 735: Add a comma after “unpublished)”
Line 751: I’d combine these first two sentences (“experiments [8,9], whereas . . .”)
Line 769: can delete comma after “patula”
Line 771: Some rewriting necessary in the sentence that started on the previous page . . . can change “canker, are” to “canker but are”
Line 817: add “the” before “pathogen”
Line 842: add a comma after “future”
Line 845: I’d change “since” to “because”
Line 852: Suggest changing “taken into account” to “considered”
Lines 853-854: Consider separating main list items with semi-colons due to commas within items
Line 863: Add a comma after “circinatum”
Lines 866-867: Suggest changing “of” to “for” and deleting “order”
Author Response
Please see attached pdf file.

Reviewer 3 Report
I find this an excellent review that was very needed since the information about F. circinatum is very disperse and not always in standard scientific literature. Thank you for this contribution.
More comments in the attached document

Author Response
Please see attached pdf file.
